# Robustness to Subpopulation Shift with Domain Label Noise via Regularized Annotation of Domains

**Nathan Stromberg**                                          *nstrombe@asu.edu*
*Arizona State University*

**Rohan Ayyagari**                                            *rayyaga2@asu.edu*
*Arizona State University*

**Monica Welfert**                                            *mwelfert@asu.edu*
*Arizona State University*

**Sanmi Koyejo**                                              *sanmi@cs.stanford.edu*
*Stanford University*

**Richard Nock**                                              *richardnock@google.com*
*Google Research*

**Lalitha Sankar**                                            *lsankar@asu.edu*
*Arizona State University*

**Reviewed on OpenReview:** `https://openreview.net/forum?id=l8E68fD6yp`

## Abstract

Existing methods for last layer retraining that aim to optimize worst-group accuracy (WGA) rely heavily on well-annotated groups in the training data. We show, both in theory and practice, that annotation-based data augmentations using either downsampling or upweighting for WGA are susceptible to domain annotation noise. The WGA gap is exacerbated in high-noise regimes for models trained with vanilla empirical risk minimization (ERM). To this end, we introduce Regularized Annotation of Domains (RAD) to train robust last layer classifiers without needing explicit domain annotations. Our results show that RAD is competitive with other recently proposed domain annotation-free techniques. Most importantly, RAD outperforms state-of-the-art annotation-reliant methods even with only 5% noise in the training data for several publicly available datasets.

## 1 Introduction

Last-layer retraining (LLR) has emerged as a method for using embeddings from pretrained models to quickly and efficiently learn classifiers in new domains or for new tasks. Because only the linear last layer is retrained, LLR allows transferring to new domains/tasks with much fewer examples than required to train a deep network from scratch. One promising use of LLR is to retrain deep models with a focus on fairness or robustness, and because data is frequently made up of distinct subpopulations (oft referred to as groups[1] which we take as a tuple of class and domain labels) (Yang et al., 2023), ensuring both fairness between subpopulations and robustness to shifts among subpopulations remains an open problem.

One way to be robust to these types of shifts and/or to be fair across groups is to optimize for the accuracy of the group that achieves the lowest accuracy, i.e., the worst-group accuracy (WGA). WGA thus presents a

---

[1]we will use these terms interchangeably

lower bound on the overall accuracy of a classifier under any subpopulation shift, thereby assuring that all groups are well classified.

State-of-the-art (SOTA) methods for optimizing WGA generally modify either the distribution of the training data (Kirichenko et al., 2023; Giannone et al., 2021; LaBonte et al., 2023) or the training loss (Arjovsky et al., 2019; Liu et al., 2021; Sagawa et al., 2020; Qiu et al., 2023) to account for imbalance amongst groups and successfully learn a classifier which is fair across groups. These methods also use some form of implicit or explicit regularization in the retraining step to limit overfitting to either the group imbalances or the spuriously correlated features in the training data. Kirichenko et al. (2023) argue that strong $\ell_1$ regularization plus data augmentation helps to learn "core features," i.e., those correlated with the label for all examples. Thus, one can instead use regularization without data augmentation to learn spurious features explicitly, in which case misclassified examples can be viewed as belonging to minority groups. This allows us to avoid the explicit use of group annotations.

We examine two representative data augmentation methods, namely downsampling (Kirichenko et al., 2023; LaBonte et al., 2023; Chaudhuri et al., 2023) and upweighting (Idrissi et al., 2022; Liu et al., 2021; Qiu et al., 2023), which achieve SOTA WGA with simple modifications to the data and loss, respectively. In the simplest setting, each of these methods requires access to correctly annotated groups to balance the contribution of each group to the loss. In practice, group annotations are often noisy (Wei et al., 2022), which can be caused by either domain noise, label noise, or both. Label noise generally affects classifier training and data augmentation, making analysis more challenging. We consider only domain noise so that we can compare to existing methods for enhancing WGA. Furthermore, focusing on domain noise presents a stepping stone to analyzing group noise in general.

## 1.1 Our Contributions

We present theoretical guarantees for the WGA under domain noise when modeling last layer representations as symmetric mixtures. We show that both DS and UW achieve identical WGA and degrade significantly with an increasing percentage of symmetric domain noise, in the limit degrading to the performance of empirical risk minimization (ERM). This is further confirmed with numerical experiments for a synthetic Gaussian mixture dataset modeling latent representations.

Our key contribution is a two-step methodology involving: (i) *regularized annotation of domains* (RAD) to pseudo-annotate examples by using a highly regularized model trained to learn spuriously correlated features. By learning the spurious correlations, RAD constructs a set of examples for which such correlations do not hold; we identify these as minority examples. (ii) LLR using all available data, while upweighting (UW) examples in the pseudo-annotated minority.

*This combined approach, denoted RAD-UW*, captures the key observation made by many that regularized LLR methods are successful as they implicitly differentiate between "core" and "spurious" features. We test RAD-UW on several large publicly available datasets and demonstrate that it achieves SOTA WGA even with noisy domain annotations. Additionally, RAD-UW incurs only a minor opportunity cost for not using domain labels even in the noise-free setting.

## 1.2 Related Works

Downsampling has been explored extensively in the literature and appears to be the most common method for achieving good WGA. Kirichenko et al. (2023) propose deep feature reweighting (DFR), which downsamples the majority groups to the size of the smallest group and then retrains the last layer with strong $\ell_1$ regularization. Chaudhuri et al. (2023) explore the effect of downsampling theoretically and show that downsampling can increase WGA under certain data distribution assumptions. LaBonte et al. (2023) use a variation on downsampling to achieve competitive WGA without domain annotations using implicitly regularized identification models.

Upweighting has been used as an alternative to downsampling as it does not require removing any data. Idrissi et al. (2022) show that upweighting relative to the proportion of groups can achieve strong WGA, and Liu et al. (2021) extend this idea (using upsampling from the same dataset, which is equivalent to

upweighting) to the domain annotation-free setting using early-stopped models. Qiu et al. (2023) use the loss of the pretrained model to upweight samples that are difficult to classify, thus circumventing the need for domain annotations.

Domain annotation-free methods generally use a secondary model to identify minority groups. Qiu et al. (2023) use the pretrained model itself but do not explicitly identify minority examples and instead upweight proportionally to the loss. Unfortunately, this ties their identification method to the choice of the loss. Liu et al. (2021) and Giannone et al. (2021) both consider fully retraining the pretrained model as opposed to only the last layer, but their method of minority identification using an early stopped model is considered in the last layer in LaBonte et al. (2023). LaBonte et al. (2023) not only consider early stopping as implicit regularization for their identification model, but also dropout (randomly dropping weights during training).

Wei et al. (2022) show that human annotation of image class labels can be noisy with up to a 40% noise proportions, thus motivating the need for domain annotation-free methods for WGA. It is likely that domain annotations are noisy with similar frequency, especially since class and domain labels are frequently interchanged, like in CelebA (Liu et al., 2015). Domain noise has not been widely considered in the WGA literature, but Oh et al. (2022) consider robustness to class label noise. They utilize predictive uncertainty from a robust identification model to select an unbiased retraining set.

**Our work** differs from SOTA methods on several fronts. First, we present a theoretical analysis of DS and UW under noise (and structured distributions for the last layer representation), which motivates our method. Secondly, we provide intuition for our RAD-UW method and the need for explicit regularization via arguments about "spurious" and "core" features à la DFR (Kirichenko et al., 2023). Finally, our method requires access only to the last layer of the pretrained model and only the final weights. This is in contrast to LaBonte et al. (2023) in which early-stopped versions of the base model are needed to get the best performance and Liu et al. (2021) in which the entire model weights are needed. With our extensive experiments, we demonstrate that downsampling methods such as those presented in Kirichenko et al. (2023) and LaBonte et al. (2023) have a significantly higher variance than upweighting methods such as RAD-UW.

## 2 Problem Setup

We consider a supervised classification setting and assume that the LLR methods have access to a representation of the *ambient* (original high-dimensional data such as images, etc.) data, the ground-truth label, as well as the (possibly noisy) domain annotation. Taken together, the label and domain combine to define the group annotation for any sample. More formally, the training dataset is a collection of i.i.d. tuples of the random variables $(X_a, Y, D) \sim P_{X_a Y D}$, where $X_a \in \mathcal{X}_a$ is the ambient high-dimensional sample, $Y \in \mathcal{Y}$ is the class label, and $D \in \mathcal{D}$ is the domain label. Here, we present the problem as generic multi-class, multi-domain learning, but for ease of analysis, we will later restrict ourselves to the binary class, binary domain setting. Since the focus here is on learning the linear last layer, we denote the *latent* representation that acts as an input to this last layer by $X := \phi(X_a)$ for an embedding function $\phi : \mathcal{X}_a \to \mathcal{X} \subseteq \mathbb{R}^m$ such that the LLR dataset is $(X, Y, D) \sim P_{XYD}$.

The tuples $(Y, D)$ of class and domain labels partition the examples into $g := |\mathcal{Y} \times \mathcal{D}|$ different groups with priors $\pi^{(y,d)} := P(Y = y, D = d)$ for $(y, d) \in \mathcal{Y} \times \mathcal{D}$. We denote the linear correction applied in the latent space of a pretrained model as $f_\theta : \mathcal{X} \to \mathbb{R}^{|\mathcal{Y}|}$, which is parameterized by a linear decision boundary $\theta = (w, b) \in \mathbb{R}^{m \times |\mathcal{Y}|} \times \mathbb{R}^{|\mathcal{Y}|}$ given by

$$f_\theta(x) = \sigma(w^T x + b). \tag{1}$$

where $\sigma : \mathbb{R}^{|\mathcal{Y}|} \to (0, 1)^{|\mathcal{Y}|}$ is the link function (e.g., softmax). The prediction of $f_\theta(x)$ is given by

$$\hat{Y} = \arg\max_i f_\theta^{(i)}(x). \tag{2}$$

A general formulation for obtaining the optimal $f_{\theta^*}$ is:

$$\theta^* = \arg\min_\theta \mathbb{E}_{P_{XYD}}[c(Y, D)\ell(f_\theta(X), Y)], \tag{3}$$

where $\ell : \mathbb{R}^{|\mathcal{Y}|} \times \mathcal{Y} \to \mathbb{R}_+$ is a loss function and $c : \mathcal{Y} \times \mathcal{D} \to \mathbb{R}$ is the per-group cost which can be used to correct for imbalances in the data (Idrissi et al., 2022) or to correct for noise in the training data (Patrini et al., 2017).

We desire a model that makes fair decisions across groups, and therefore, we evaluate worst-group accuracy, i.e., the minimum accuracy among all groups, defined for a model $f_\theta$ as

$$\mathrm{WGA}(f_\theta) := \min_{(y,d) \in \mathcal{Y} \times \mathcal{D}} A^{(y,d)}(f_\theta), \tag{4}$$

where $A^{(y,d)}(f_\theta)$ denotes the per-group accuracy for the group $(y,d) \in \mathcal{Y} \times \mathcal{D}$,

$$A^{(y,d)}(f_\theta) := P_{X|YD}(\hat{Y} = Y | Y = y, D = d), \tag{5}$$

where $\hat{Y}$ is calculated as in (2).

## 2.1 Data Augmentation

**Downsampling** (DS) reduces the number of examples in majority groups such that minority and majority groups have the same sample size. In practice, this reduces the dataset size from $n$ to $g \times n_{min}$ where $n_{min}$ is the number of examples in the smallest group. In the population setting (as $n \to \infty$ and $n_{min} \approx \pi_{min} \times n$), this is equivalent to setting all group priors equal to $1/g$.

**Upweighting** (UW) does not remove data but weights the loss more for minority examples and less for majority samples. Generally, the upweighting factor $c$ can be a hyperparameter, though often one uses the inverse of the prevalence of the group in practice, which estimates

$$c(y,d) = \frac{1}{g\pi^{(y,d)}}. \tag{6}$$

This selection is motivated by the minimization problem in (3), where this choice of $c$ allows the optimization to happen independently of the group priors. This is explored more in Proposition 3.1.

## 2.2 Domain Noise

We model noise in the domain label as symmetric label noise (SLN) with probability (w.p.) $p$. That is, for a sample $(X, Y, D) \sim P_{XYD}$, we do not observe $D$ directly but $D$ w.p. $1 - p$ and $\bar{D}$ w.p. $p$ where $\bar{D}$ is drawn uniformly at random from $\mathcal{D} \setminus \{D\}$. This is equivalent to flipping $D$ w.p. $p$ in the binary domain setting.

In practice, while the training data is usually regarded as noisy, it is frequently necessary to have a small holdout that is clean and fully annotated. This allows for hyperparameter selection without being affected by noisy annotations and aligns with domain annotation-free settings which generally have a labeled holdout set (Liu et al., 2021; Giannone et al., 2021; LaBonte et al., 2023).

# 3 Theoretical Guarantees

We first consider the general setting (multi-class, multi-domain) and show that the models learned after downsampling ($\theta^*_{\mathrm{DS}}$) and upweighting ($\theta^*_{\mathrm{UW}}$) are the same in the population setting.

**Proposition 3.1.** *For any given $P_{XYD}$ and loss $\ell$, the objectives in (3), when modified appropriately for DS and UW, are the same. Therefore, if a minimizer exists for one it also exists for the other, i.e., $\theta^*_{DS} = \theta^*_{UW}$.*

*Proof.* The key idea of the proof is that the upweighting factor is proportional to the inverse of the priors on each group. Thus, for any $f_\theta$ and $P_{XYD}$, the expected loss is

$$\mathbb{E}_{X,Y,D}[\ell(f_\theta(X), Y) c(Y, D)] \tag{7}$$

and can be decomposed into an expectation over groups. For such a decomposition, the priors from the expected loss cancel with the upweighting factor and we recover the downsampled problem with uniform priors.

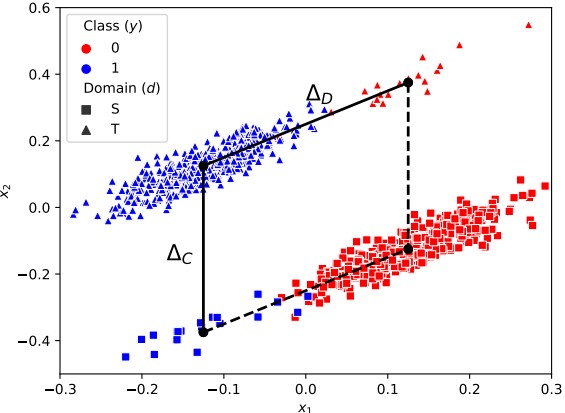 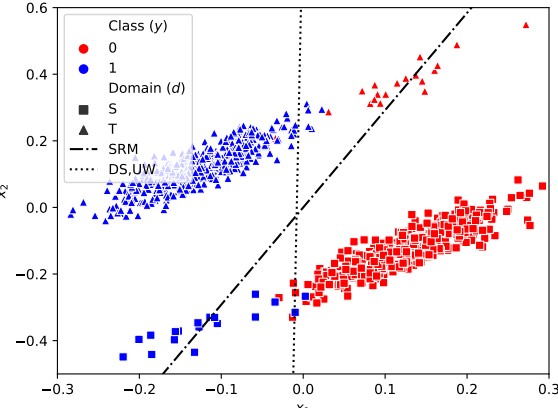

Figure 1: Sample drawn from a distribution satisfying Assumptions 3.3 to 3.6. $\Delta_C$ and $\Delta_D$ are shown as line segments between means. Additionally the classifiers learned by SRM, DS, and UW are shown. It is clear that DS and UW learn the separator which is unaffected by spurious correlation.

We now consider the setting where each group, given by a tuple of binary domain ($\mathcal{D} = \{S, T\}$) and binary class labels ($\mathcal{Y} = \{0, 1\}$), is symmetrically distributed with different means but equal covariance. We additionally impose a structural condition studied in Yao et al. (2022) which allows us to theoretically analyze the weights and performance of least-squares-type algorithms, learning a simplified linear classifier with squared loss.

**Definition 3.2.** () A probability distribution $Q$ on $\mathbb{R}$ is *symmetric* around its mean $\mu$ if its CDF $\Phi$ satisfies

$$1 - \Phi(x - \mu) = \Phi(-x + \mu), \quad \forall x \in \mathbb{R}.$$

**Assumption 3.3.** Conditioned on $Y$ and $D$, $X \in \mathcal{X}$ is distributed according to $P_{X|y,d}$ for $(y, d) \in \mathcal{Y} \times \mathcal{D}$, with mean $\mu^{(y,d)} \coloneqq \mathbb{E}[X|Y = y, D = d] \in \mathbb{R}^m$ and positive semidefinite covariance $\Sigma \in \mathbb{R}^{m \times m}$ such that the distribution of $a^T X | (Y = y, D = d)$ is *symmetric* for any $a \in \mathbb{R}^m$ and its CDF $\Phi_{a^T X|y,d} : \mathbb{R} \to [0, 1]$ is strictly increasing. Additionally, we place priors $\pi^{(y,d)}$, $(y, d) \in \mathcal{Y} \times \mathcal{D}$, on each group and priors $\pi^{(y)} \coloneqq P(Y = y)$, $y \in \mathcal{Y}$, on each class.

Note that Assumption 3.3 holds for many conditional distributions on $X$ including spherically symmetric distributions (see Fang (2018, Theorem 2.4) along with the fact that spherically symmetric distributions have symmetric marginals), Guassian distributions, and distributions with sub-independent and symmetric marginals. For ease of intuition, we focus mainly on Guassian groups. Following Chaudhuri et al. (2023) we deal only with distributions which induce strictly increasing CDFs so that there is an explicit tradeoff between majority and minority group performance.

**Assumption 3.4.** The minority groups have equal priors, i.e., for $\pi_0 \leq 1/4$,

$$\pi^{(0,R)} = \pi^{(1,B)} = \pi_0 \text{ and } \pi^{(1,R)} = \pi^{(0,B)} = 1/2 - \pi_0.$$

Also, the class priors are equal, i.e., $\pi^{(0)} = \pi^{(1)} = 1/2$.

**Assumption 3.5.** The difference in means between classes in a domain $\Delta_D \coloneqq \mu^{(1,d)} - \mu^{(0,d)}$ is constant for $d \in \mathcal{D}$.

Assumption 3.5 also implies that the difference in means between domains within the same class $\Delta_C \coloneqq \mu^{(y,B)} - \mu^{(y,R)}$ is also constant for each $y \in \mathcal{Y}$. We see this by noting that each group mean makes up the vertex of a parallelogram. This is illustrated in Figure 1, where $\Delta_D$ and $\Delta_C$ are shown on data samples drawn from a distribution satisfying Assumptions 3.3 to 3.5.

**Assumption 3.6.** $\Delta_D$ and $\Delta_C$ are orthogonal with respect to $\Sigma^{-1}$, i.e., $\Delta_C^T \Sigma^{-1} \Delta_D = 0$.

We see an example of a dataset drawn from a distribution satisfying Assumptions 3.3 to 3.6 in Figure 1. The data is generated as a mixutre of 2D Gaussians with parameters,

$$\Delta_C = \begin{pmatrix} 0 & -0.5 \end{pmatrix}^T \qquad\qquad \Delta_D = \begin{pmatrix} -0.25 & -0.25 \end{pmatrix}^T$$

$$\Sigma = \begin{pmatrix} .003 & .003 \\ .003 & .004 \end{pmatrix} \qquad\qquad \pi_0 = \frac{1}{50}.$$

While this is a simplified view of binary class, binary domain latent groups, these tractable assumptions allow us to make theoretical guarantees about the performance of downsampling and upweighting under noise. We see that the general trends observed in this simplified setting hold in large publicly available datasets in Section 5.4.

We now show that upweighting and downsampling achieve identical worst-group accuracy in the population setting (i.e., infinite samples) and both degrade with SLN in the domain annotation while ERM is unaffected by domain noise.

**Theorem 3.7.** *Consider the model of latent groups satisfying Assumptions 3.3 to 3.6 under symmetric domain label noise with parameter $p$. Let $f_\theta(x) = w^T x + b$ and $\hat{Y} = \mathbb{1}\{f_\theta(x) > 1/2\}$. In this setting, let $\theta_{UW}^{(p)}$ and $\theta_{DS}^{(p)}$ denote the solution to (3) under squared loss for UW and DS, respectively. For any $\pi_0 \in (0, 1/4]$, the WGA of both augmentation approaches are equal and degrade smoothly in $p \in [0, 1/2]$ to the baseline WGA of (3) with no augmentation (with optimal parameter $\theta_{ERM}$). That is,*

$$WGA(\theta_{ERM}) \le WGA(\theta_{DS}^{(p)}) = WGA(\theta_{UW}^{(p)})$$

*with equality at $p = 1/2$ or $\pi_0 = 1/4$.*

*Proof Sketch.* The proof of Theorem 3.7 is presented in Appendix A and involves showing that the WGA for downsampling under domain label noise is the same as that for ERM (which is noise agnostic) but with a different prior dependent on $p$. Our proof refines the analysis in Yao et al. (2022) and involves the noisy prior.

Fundamentally, this result can be seen as an effect of the domain noise on the perceived (noisy) priors $\pi_0^{(p)}$ of the minority groups, which can be derived as

$$\pi_0^{(p)} := (1-p)\pi_0 + p\left(1/2 - \pi_0\right). \tag{8}$$

As $p \to 1/2$ in (8), the minority prior *perceived* by both UW and DS tends to a balanced prior across groups. A UW augmentation thus would weight in inverse proportion to the corresponding noisy (and not the true) prior for each group. The *true prior* of the minority group after DS can be derived as (see Appendix A)

$$\pi_{\text{DS}}^{(p)} := \frac{(1-p)\pi_0}{4\pi_0^{(p)}} + \frac{p\pi_0}{4(1/2 - \pi_0^{(p)})}. \tag{9}$$

Thus, with noisy domain labels, instead of the desired balanced group priors after downsampling, from (9), DS results in a true minority prior that decreases from $1/4$ to $\pi_0$. Thus, noise in domain labels drives inaction from both augmented methods as $p$ increases.

We see the effect of noise in a numerical example in Figure 2, noting that while the theorem is in the population setting, our numerical example uses finite sample methods with $n = 10,000$. This shows that the performance of each method quickly degrades even in this simple setting. The ERM performance, however, remains constant because ERM does not use domain information. This motivates us to examine a robust method that does not use domain information at all but is more effective than ERM in terms of WGA.

## 4   Regularized Annotation of Domains

When domain annotations are unavailable (or noisy), LaBonte et al. (2023) use implicitly regularized models trained on the imbalanced retraining data to annotate the data. We take a similar approach but explicitly

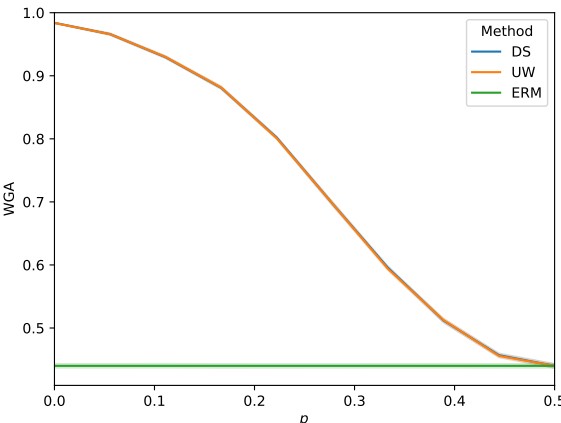

Figure 2: For latent Gaussian data, the WGA of DS and UW (seen as overlapping) decreases as the noise prevalence $p$ increases to $1/2$. At the extreme point, the WGA of ERM is recovered.

regularize our pseudo-annotation model with an $\ell_1$ penalty. The intuition behind using an $\ell_1$ penalty is similar to that of DFR (Kirichenko et al., 2023). Where Kirichenko et al. (2023) argue that an $\ell_1$ penalty helps to select only "core features" when trained on a group-balanced dataset, we argue that the same penalty with a large multiplicative factor will help to select spuriously correlated features when trained on the original imbalanced data. If we can successfully learn the spuriously correlated features, those samples which are correctly classified can be viewed as majority samples (those for which the spurious correlation holds) and those which are misclassified can be seen as minority samples.

We introduce RAD (Regularized Annotation of Domains) which uses a highly $\ell_1$ regularized linear model to pseudo-annotate domain information by quantizing true domains to binary majority and minority annotations. Pseudocode for this algorithm is presented in Algorithm 1. We see that a strongly regularized classified is first learned, $f_{\text{ID}}$, which allows us to identify minority points through misclassification. We use these annotations, $\tilde{d}$, to upweight examples in the next step

**RAD-UW** (Algorithm 2) involves learning sequentially: (i) a pseudo-annotation RAD model (outlined in Algorithm 1), and (ii) a regularized linear retraining model, which is trained on all examples while upweighting the pseudo-annotated minority examples output by RAD, as the solution $\hat{\theta}^*_{\text{RAD-UW}}$ optimizing the empirical version of (3) using the logistic loss $\ell_L$ with $\ell_1$ regularization as:

$$\hat{\theta}^*_{\text{RAD-UW}} = \arg\min_{\theta} \frac{1}{n} \sum_{i=1}^{n} c(\tilde{d}_i)\ell_L(f_\theta(x_i), y_i) + \lambda\|w\|_1. \tag{10}$$

Here again, the $\ell_1$ regularization comes into play. While the regularization in the first step encouraged the biased model to misclassify minority points, here the same regularization encourages learning features which do well both for the majority and (upweighted) minority. This discourages learning spurious features which may be present (and helpful) in the majorities.

**RAD-UW vs. M-SELF**  While both RAD and SELF are two-stage methods which utilize the biases of an "identification" model to pseudoannotate points in minority classes, RAD explicitly trains such a biased classifier using a strong $\ell_1$ penalty whereas the classifier used by SELF is the pretrained classification head. The benefit of retraining the classifier is that we can force the model to rely more strongly on spurious correlations. We see in practice that this leads to drastic increases in performance on some datasets. A side effect of this retraining is that we can more effectively correct models which were originally trained with noisy data, as the noise affects the classification head more strongly than the embeddings themselves. We explore this is Appendix G.

**Algorithm 1** Regularized Annotation of Domains (RAD)

---

**Input:** data $D = (x, y)$
Train classifier $f_{\text{ID}}$ on $D$ with $\ell_1$ factor $\lambda_{\text{ID}}$
**for** $(x_i, y_i) \in D$ **do**
  **if** $f_{\text{ID}}(x_i) \neq y_i$ **then**
    $\tilde{d}_i \leftarrow 1$
  **else**
    $\tilde{d}_i \leftarrow 0$
  **end if**
**end for**
**Return:** $(x, y, \tilde{d})$, the pseudo-annotated data

**Algorithm 2** RAD-UW

---

**Input:** data $D = (x, y)$
$(x, y, \tilde{d}) \leftarrow \text{RAD}(x, y)$
Train classifier $f_{\text{retrain}}$ on $(x, y)$ while upweighting examples where $\tilde{d} = 1$ by factor $c$
**Return:** $\hat{y} = f_{\text{retrain}}(x)$, the estimated class label of $x$

## 5 Empirical Results

We present worst-group accuracies for several representative methods across four large publicly available datasets. Note that for all datasets, we use the training split to train the embedding model. Following prior work (Kirichenko et al., 2023; LaBonte et al., 2023), we use half of the validation as retraining data, i.e., training data for only the last layer, and half as a clean holdout.

### 5.1 Datasets

**CMNIST** (Arjovsky et al., 2019) is a variant of the MNIST handwritten digit dataset in which digits 0-4 are labeled $y = 0$ and digits 5-9 are labeled $y = 1$. Further, 90% of digits labeled $y = 0$ are colored green and 10% are colored red. The reverse is true for those labeled $y = 1$. Thus, we can view color as a domain and we see that the color of the digit and its label are correlated.

**CelebA** (Liu et al., 2015) is a dataset of celebrity faces. For this data, we predict hair color as either blonde ($y = 1$) or non-blonde ($y = 0$) and use gender, either male ($d = 1$) or female ($d = 0$), as the domain label. There is a natural correlation in the dataset between hair color and gender because of the prevalence of blonde female celebrities.

**Waterbirds** (Sagawa et al., 2020) is a semi-synthetic dataset which places images of land birds ($y = 1$) or sea birds ($y = 0$) on land ($d = 1$) or sea ($d = 0$) backgrounds. There is a correlation between background and the type of bird in the training data but this correlation is removed in the validation data.

**MultiNLI** (Williams et al., 2018) is a text corpus dataset widely used in natural language inference tasks. For our setup, we use MultiNLI as first introduced in (Oren et al., 2019). Given two sentences, a premise and a hypothesis, our task is to predict whether the hypothesis is either entailed by, contradicted by, or neutral with the premise. There is a spurious correlation between there being a contradiction between the hypothesis and the premise and the presence of a negation word (no, never, etc.) in the hypothesis.

**CivilComments** (Borkan et al., 2019) is a text corpus dataset of public comments on news websites. Comments are labeled either as toxic ($y = 1$) or civil ($y = 0$) and the spurious attribute is the presence ($d = 1$) or absence ($d = 0$) of a minority identifier (e.g. LGBTQ, race, gender). There is a strong class imbalance (most comments are civil), though the domain imbalance is modest.

### 5.2 Importance of $\ell_1$ Regularization

We emphasize that a strong $\ell_1$ regularizer is critical to the success of our method through the intuition of DFR, but it remains to be seen how this bears out in practice. To this end, we examine combinations of both $\ell_1$ and the common $\ell_2$ penalties. We see in Table 1 that the retraining regularizer has only a small impact on the overall performance, while the identification regularization has more impact. We note that for CelebA, a

Table 1

| Regularizer | | Dataset WGA | | | |
|---|---|---|---|---|---|
| ID | Retrain | CMNIST | WB | CelebA | Civil Comments |
| $\ell_1$ | $\ell_1$ | **94.06** | **92.52** | **83.51** | **81.57** |
| $\ell_1$ | $\ell_2$ | 93.89 | 89.53 | 82.59 | 74.97 |
| $\ell_2$ | $\ell_1$ | 94.04 | 92.18 | 68.39 | 51.17 |
| $\ell_2$ | $\ell_2$ | 93.86 | 90.98 | 64.28 | 53.00 |

real-world image dataset of faces, $\ell_1$ regularization shows dramatic gains, while the synthetic Waterbirds and CMNIST datasets are much closer.

## 5.3 Main Results

We present results for both group- and class-only-dependent methods using both downsampling and upweighting. Additionally, we present results for vanilla LLR, which performs no data or loss augmentation step before retraining. Finally, we present results for two-stage last layer methods, namely misclassification SELF (M-SELF) and RAD with upweighting (RAD-UW). Every retraining method solves the following empirical optimization problem:

$$\hat{\theta}^* = \arg\min_\theta \frac{1}{n}\sum_{i=1}^{n} c(d_i, y_i)\ell(f_\theta(x_i), y_i) + \lambda\|w\|_1, \tag{11}$$

which can be seen as a finite sample version of (3) with an $\ell_1$ regularization. For $\ell$, we use the logistic loss.

The group-dependent downsampling procedure we have adopted is the same as that of DFR, introduced in Kirichenko et al. (2023), but the DFR methodology averages the learned model over 10 training runs. This could help to reduce the variance, but we do not implement this so as to directly compare different data augmentation methods since most others do not so either. More generally, we could apply model averaging to any of the data augmentation methods. We explore the effect of model averaging in Appendix E.

We use the logistic regression implementation from the `scikit-learn` (Pedregosa et al., 2011) package for the retraining step for all presented methods. For all final retraining steps (including LLR) an $\ell_1$ regularization is added. The strength of the regularization $\lambda$ is a hyperparameter selected using the clean holdout. The upweighting factor for group annotation-inclusive UW methods is given by the inverse of the perceived prevalence for each group or class.

For our RAD-UW method, we tune the regularization strength $\lambda_{\text{ID}}$ for the pseudo-annotation model via a grid search. We additionally tune the regularization strength $\lambda$ of the retraining model along with the upweighting factor $c$, which is left as a hyperparameter because the identification of domains by RAD is only binary and may not reflect the domains in the clean holdout data. The same upweighting factor is used for every pseudo-annotated minority sample.

Note that for all of the WGA results, we report the mean WGA over 10 independent noise seeds and 10 training runs for each noise seed. We also present the standard deviation around this mean, which will reflect the variance over training runs with the optimal hyperparameters. For misclassification SELF (M-SELF) (LaBonte et al., 2023), we present both our own implementation of their algorithm and their results directly. It should be noted that they report the mean and standard deviation over only three independent runs and without noise; for the noisy setting, we observe that their method does not use domain annotations and so is unaffected by domain noise just as RAD-UW is. Our implementation utilizes the same embeddings as RAD-UW so we can ensure any differences are artifacts of the algorithms rather than the upstream model. The importance of the pretrained model is an important and underexplored factor in the literature which we leave as future work.

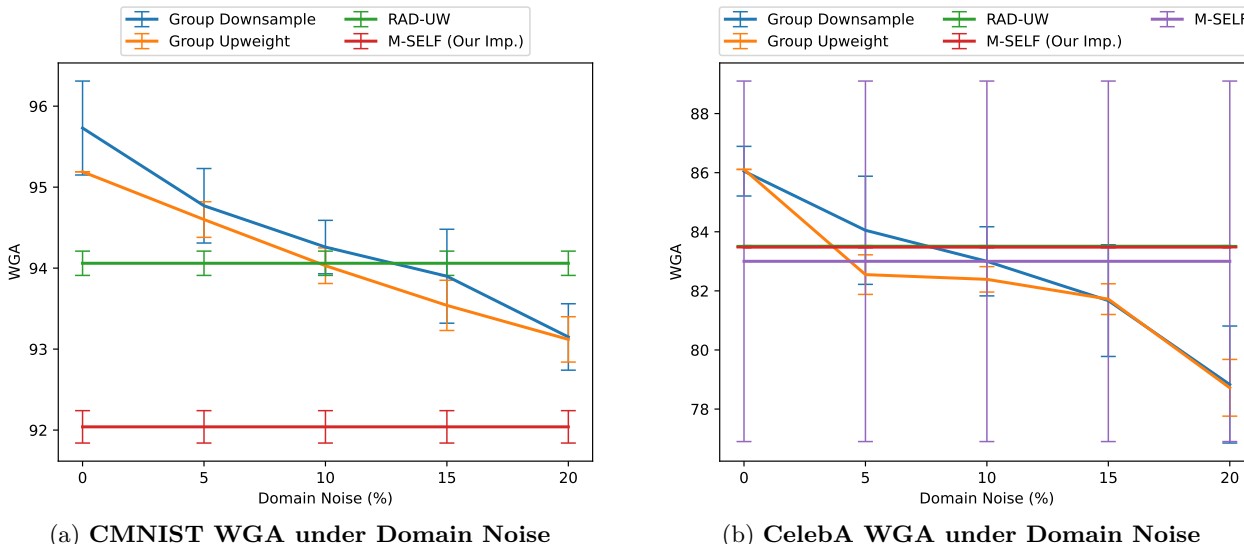

(a) **CMNIST WGA under Domain Noise**  (b) **CelebA WGA under Domain Noise**

Figure 3: Domain-dependent methods, group downsampling and upweighting, decline in performance as the domain noise increases. Meanwhile, RAD-UW remains consistent and matches or outperforms these methods starting at 10% domain noise. The high variance of M-SELF in CelebA is concerning and is likely due to the class balancing performed by M-SELF.

## 5.4 Worst-Group Accuracy under Noise

We now detail the results on all datasets in Figures 3a, 3b, 4a, 4b and 5a. See Appendix F for tables including class dependent versions of downsampling and upweighting. We choose not to include them here as their performance is so poor as to distract from the results.

For CMNIST, we present results in Figure 3a and Table 8; we see that each method that uses domain information achieves strong WGA at 0% domain annotation noise, but for increasing noise, their performance drops noticeably. Additionally the methods which rely only on class labels without inferring domain membership are consistent across noise levels, but are outdone by their domain-dependent counterparts even at 20% noise. Finally, RAD-UW achieves WGA comparable with the group-dependent methods, and surpasses their performance after 10% noise in domain annotation.

The results for CelebA are presented in Figure 3b and Table 9. We first note that LLR achieves significantly lower WGA than any other method owing to strong spurious correlation even in the retraining data. We also note that every domain-dependent method has a much more significant decline in performance for CelebA than for CMNIST. Here, RAD-UW outperforms the domain-dependent methods even at 10% SLN. Not only this, but RAD beats the performance of SELF (LaBonte et al., 2023) for this dataset with much lower variance.

For Waterbirds, we see in Figure 4a and Table 10 that noise, even significant amounts of it, has little effect on any of the methods considered. This is because the existing splits have a domain-balanced validation, which we use here for retraining. Thus domain noise does not affect the group priors at all as argued analytically in (8) with $\pi_0 = 1/4$. Even so, we see that RAD-UW is competitive with existing methods, including the domain annotation-free methods of LaBonte et al. (2023).

For MultiNLI, we see in Figure 4b and Table 11 that RAD-UW is able to match domain-dependent methods at 5% noise and outperform them at 10%, but is beaten by M-SELF's reported result. We note that we were not able to replicate their performance with our implementation. This may be because MultiNLI has weaker spurious correlation than the other datasets we examine. Also important to note is the effect of noise on MultiNLI. For 5% and 10% noise levels, we see a dramatic drop in WGA for domain-dependent methods, but this levels off as we recover the WGA of LLR. Thus even for 10% noise, LLR is competitive with top domain-dependent methods.

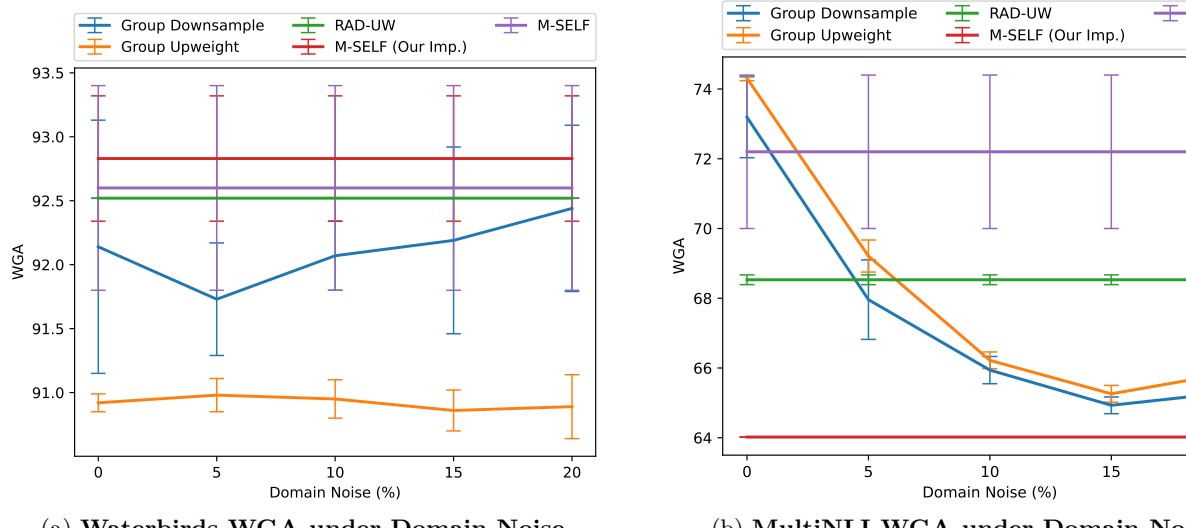

(a) **Waterbirds WGA under Domain Noise**    (b) **MultiNLI WGA under Domain Noise**

Figure 4: We see that RAD-UW and M-SELF strongly outperform domain-dependent methods for most noise levels. Waterbirds is domain balanced from the beginning, so adding domain noise does not strongly bias the downsampling or upweighting classifiers.

Finally, for CivilComments, we that in Figure 5a and Table 12 that RAD-UW drastically outperforms M-SELF and is competitive with domain-dependent methods. Because CivilComments is generally domain balanced, domain noise does not have a large effect on the performance of GDS and GUW. On the other hand, the extreme class imbalance of the dataset leads to poor performance by M-SELF. The robustness of RAD-UW to extreme class imbalnce is an important factor in its success and likely due to the downsampling of competing methods.

An interesting observation is that for almost all datasets and noise levels, the variance of group-dependent downsampling is consistently larger than upweighting at the same noise levels. This behavior is likely caused by two key issues: (i) DS reduces the dataset size, and (ii) it randomly subsamples the majority, each of which could increase the variance of the resulting classifier. DFR (Kirichenko et al., 2023) has attempted to remedy this issue by averaging the linear model that is learned over 10 different random downsamplings, but this increases the complexity of learning the final model. Our results raise questions on the prevalence of downsampling over the highly competitive upweighting method in the setting of WGA.

A similar comment can be made with respect to RAD-UW and M-SELF. While M-SELF has strong performance on several datasets, it consistently has higher variance than RAD-UW and struggles on datasets with large class imbalance. This is likely due to the class balancing that LaBonte et al. (2023) use in the retraining step. RAD-UW avoids this issue by using all of the available data and upweighting error points.

Finally, we consider the opportunity cost of ignoring domain annotations in the worst case, i.e., the domain labels we have are in fact noise-free but we still ignore them. In Figure 5b, we see that the domain annotation-free methods cost very little in terms of lost WGA when training on cleanly annotated data. This loss is minimal in comparison to the cost of training using an annotation-dependent method when the domain information is noisy. Thus, if there are concerns about domain annotation noise in the training data, it is safest to use a domain annotation-free approach.

## 6    Discussion

We see in our experiments that these two archetypal data augmentation techniques, downsampling and upweighting, achieve very similar worst-group accuracy and degrade similarly with noise. This falls in line with our theoretical analysis, and suggests that simple symmetric mixture models for subpopulations can provide intuition for the performance of data-augmented last layer retraining methods on real datasets.

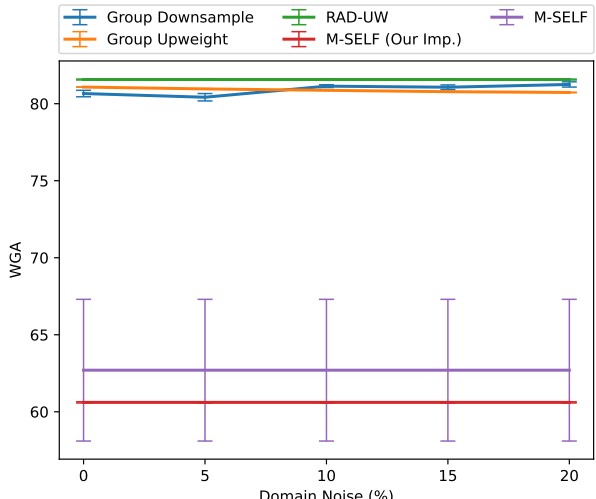 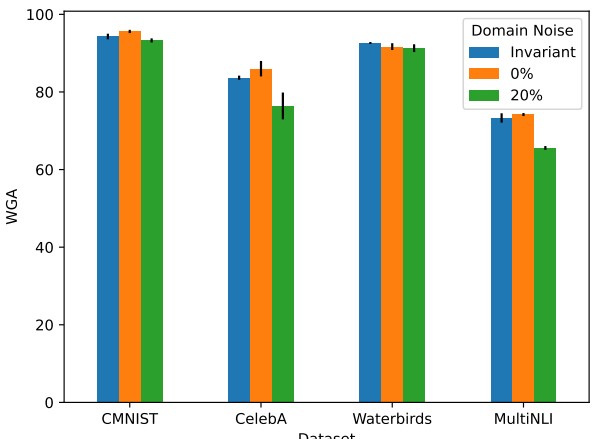

(a) **CivilComments WGA under Domain Noise**. We see that RAD-UW strongly outperforms M-SELF on this datasets, likely because of the extreme class imbalance present in the retraining set. This, along with the class balancing performed by M-SELF, results in a very small error set to retrain on. Domain dependent methods are not strongly affected by domain noise because of the relative lack of domain imbalance.

(b) **The cost of ignoring domain annotations.** For each of the datasets, we compare the WGA of the best domain-dependent method at 0% and 20% domain noise with the best WGA amongst the methods which do not use domain information (labeled "Invariant"). The cost of ignoring the domain information is very small for most datasets and in Waterbirds, utilizing domain information hurts performance somewhat. Thus, the opportunity cost of not using domain information is relatively small, while the cost of using a domain-dependent method with noise is quite large.

Our experiments also indicate that downsampling induces a higher variance in WGA, especially in the noisy domain annotation setting. While this phenomenon is not captured in our population analysis, intuitively one should expect that having a smaller dataset should increase the variance. Additionally, the models learned by group downsampling suffer from an additional dependence on the data that is selected in downsampling, which in turn increases the WGA variance.

Overall, we demonstrate that achieving SOTA worst-group accuracy is strongly dependent on the quality of the domain annotations on large publicly available datasets. Our experiments consistently show that in order to be robust to noise in the domain annotations, it is necessary to ignore them altogether. To this end, our novel domain annotation-free method, RAD-UW, assures WGA values competitive with annotation-inclusive methods. RAD-UW does so by pseudo-annotating with a highly regularized model that allows discriminating between samples with spurious features and those without and retraining on the latter with upweighting. Our comparison of RAD-UW to two existing domain annotation-free methods and several domain annotation-dependent methods clearly highlights that it can outperform existing methods even for only 5% domain label noise.

There is a significant breadth of future work available in this area. In the domain annotation-free setting, there is a gap in the literature regarding identification of subpopulations beyond the binary "minority," or "majority" groups. Identifying individual groups could help to increase the performance of retrained models and give better insight into which groups are negatively affected by vanilla LLR. Additionally, tuning hyperparameters without a clean holdout set remains an open question.

Beyond this, group noise could be driven by class label noise alongside domain annotation noise. The combination of these two types of noise would necessitate a robust loss function when training both the pseudo-annotation and retraining models, or a new approach entirely. We are optimistic that the approach presented here could be combined in a modular fashion with a robust loss to achieve robustness to more general group noise.

**Broader Impacts and Limitations**

We attempt to address the issue of subgroup fairness and robustness to subpopulation shift through a pseudo annotation strategy in the domain. This method could allow practitioners to more easily adapt existing models while assuring fair classification across subpopulations. However, because there is a not a formal guarantee of fairness, there is a possible societal consequence of practitioners assuming the retrained model is fair without verifying it. Similar lack of guarantees is a downfall of most methods in the WGA literature.

Additionally Oh et al. (2022) has demonstrated that most two-stage methods, including the full-retraining precursors to RAD and SELF, can fair dramatically with small amounts of class label noise. Addressing this issue is left as future work.

Finally we note that achieving this fairness without domain-annotation presents an opportunity when domain labels may be private. This tradeoff between privacy and fairness is an important area of research, as discussed in King et al. (2023). Perhaps there is a regime with limited private information, but enough to achieve some gains over domain-agnostic methods such as RAD and SELF.

**Acknowledgements**

Nathan Stromberg, Rohan Ayyagari, Monica Welfert, and Lalitha Sankar acknowledge support by NSF CIF-2007688, SCH-2205080, PIPP-2200161, and a Google AI for Social Good grant. Nathan Stromberg and Monica Welfert are grateful for support from Arizona State University's Dean's Fellowship. Sanmi Koyejo acknowledges support by NSF Career Award 2046795 and SCH-2205329, Stanford HAI, and Google Inc.

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

# A Proof of Theorem 3.7

Our proof can be outlined as involving five steps; these steps rely on Proposition 3.1 and include three new lemmas. We enumerate the steps below:

1. We first show in Lemma A.1 that ERM is agnostic to domain label noise.

2. We next show in Lemma A.2 that for clean data with any minority prior $\pi_0$, the WGA for ERM is given by (12)

3. In Lemma A.5 we show that the model learned after downsampling with noisy domain labels is equivalent to a clean ERM model learned with prior

$$\frac{(1-p)\pi_0}{4\pi_0^{(p)}} + \frac{p\pi_0}{4\left(1/2 - \pi_0^{(p)}\right)}.$$

4. We then show that the WGA of downsampling strictly decreases in $p$ by examining the derivative.

5. Finally we note that by Proposition 3.1, upweighting must learn the same model as downsampling

We present the three lemmas below and use them to complete the proof.

**Lemma A.1.** *ERM with no data augmentation is agnostic to the domain label noise $p$, i.e., the model learned by ERM in* (3) *in the setting of domain label noise is the same as that learned in the setting of clean domain labels (no noise).*

*Proof.* Since ERM with no data augmentation does not use domain label information when learning a model, the model will remain unchanged under domain label noise. □

In the following, let $\Phi : \mathbb{R} \to [0,1]$ be the CDF of $[w^T(X|Y,D) - \mu^{(Y,D)}]/\sqrt{w^T\Sigma w}$ for any $Y \in \mathcal{Y}$ and $D \in \mathcal{D}$

**Lemma A.2.** *Let $\theta_{ERM}$ denote the optimal model parameter learned by ERM in* (3) *using $f_\theta(x) = w^T x + b$ and $\hat{Y} = \mathbb{1}\{f_\theta(x) > 1/2\}$. Under Assumptions 3.3 to 3.6,*

$$WGA(\theta_{ERM}) = \Phi\left(\frac{\|\Delta_D\|^2 - \tilde{c}_{\pi_0}\|\Delta_C\|^2}{2(\|\Delta_D\| + \tilde{c}_{\pi_0}\|\Delta_C\|)}\right), \tag{12}$$

*where $\tilde{c}_{\pi_0} \coloneqq (1 - 4\pi_0)/(1 + 2\pi_0(1 - 2\pi_0)\|\Delta_C\|^2)$ and $\|v\| \coloneqq \sqrt{v^T\Sigma^{-1}v}$.*

*Proof.* Since the model learned by ERM with no data augmentation is invariant to domain label noise by Lemma A.1, we derive the the general form for the WGA of a model $f_\theta$ under the assumption of having clean domain label, and therefore using the original data parameters. We begin by deriving the individual accuracy terms $A^{(y,d)}(f_\theta)$, $(y,d) \in \{0,1\} \times \{R,B\}$, with $f_\theta(x) = w^T x + b$ and $\hat{Y} = \mathbb{1}\{f_\theta(x) > 1/2\}$, as follows:

$$
\begin{aligned}
A^{(1,d)}(f_\theta) &\coloneqq P\left(\mathbb{1}\left\{w^T X + b > 1/2\right\} = Y \mid Y = 1, D = d\right) \\
&= P\left(w^T X + b > 1/2 \mid Y = 1, D = d\right) \\
&= 1 - \Phi\left(\frac{1/2 - (w^T\mu^{(1,d)} + b)}{\sqrt{w^T\Sigma w}}\right) \\
&= \Phi\left(\frac{w^T\mu^{(1,d)} + b - 1/2}{\sqrt{w^T\Sigma w}}\right),
\end{aligned}
$$

$$
\begin{aligned}
A^{(0,d)}(f_\theta) &\coloneqq P\left(\mathbb{1}\left\{w^T X + b > 1/2\right\} = Y \mid Y = 0, D = d\right) \\
&= P\left(w^T X + b \le 1/2 \mid Y = 0, D = d\right) \\
&= \Phi\left(\frac{1/2 - (w^T\mu^{(0,d)} + b)}{\sqrt{w^T\Sigma w}}\right).
\end{aligned}
$$

We now derive the optimal model parameters for ERM for any $\pi_0 \leq 1/4$. In the case of ERM, $c(y, d) = 1$ for $(y, d) \in \mathcal{Y} \times \mathcal{D}$, so the optimal solution to (3) with $f_\theta(x) = w^T x + b$ and $\hat{Y} = \mathbb{1}\{f_\theta(x) > 1/2\}$ is

$$w_{\text{ERM}} = \text{Var}(X)^{-1}\text{Cov}(X, Y), \quad \text{and} \quad b_{\text{ERM}} = \mathbb{E}[Y] - w^T \mathbb{E}[X]. \tag{13}$$

Note that

$$\mathbb{E}[Y] = \frac{1}{2},$$
$$\mathbb{E}[X] = \mathbb{E}\left[\mathbb{E}[X|D, Y]\right]$$
$$= \sum_{(y,d) \in \{0,1\} \times \{R,B\}} [\mathbb{1}(d = R)(\mu^{(0,R)} - \mu^{(0,B)}) + (1 - y)\mu^{(0,B)} + y\mu^{(1,B)}]\pi^{(y,d)}$$
$$= \frac{1}{2}(\mu^{(0,R)} + \mu^{(1,B)}).$$

Let $\pi^{(d|y)} := P(D = d|Y = y)$ for $(y, d) \in \mathcal{Y} \times \mathcal{D}$, $\mu^{(y)} := \mathbb{E}[X|Y = y]$ for $y \in \mathcal{Y}$ and $\bar{\Delta} := \mu^{(1)} - \mu^{(0)}$. We compute $\text{Var}(X)$ as follows:

$$\text{Var}(X) = \mathbb{E}[\text{Var}(X|Y)] + \text{Var}(\mathbb{E}[X|Y]) \tag{14}$$
$$= \mathbb{E}\left[\mathbb{E}[\text{Var}(X|Y, D)|Y] + \text{Var}(\mathbb{E}[X|Y, D]|Y)\right] + \text{Var}(\mathbb{E}[X|Y]) \tag{15}$$
$$= \Sigma + \mathbb{E}[\text{Var}(\mathbb{E}[X|Y, D]|Y)] + \text{Var}(\mathbb{E}[X|Y]) \tag{16}$$
$$= \Sigma + \mathbb{E}[\text{Var}(\mathbb{1}(D = R)(\mu^{(0,R)} - \mu^{(0,B)}) + (1 - Y)\mu^{(0,B)} + Y\mu^{(1,B)}|Y)] + \text{Var}(\mathbb{E}[X|Y]) \tag{17}$$
$$= \Sigma + \Delta_C \Delta_C^T \mathbb{E}[\text{Var}(\mathbb{1}(D = R)|Y)] + \text{Var}(Y(\mu^{(1)} - \mu^{(0)}) + \mu^{(0)}) \tag{18}$$
$$= \Sigma + \Delta_C \Delta_C^T \mathbb{E}[\text{Var}(D|Y)] + \bar{\Delta}\bar{\Delta}^T \text{Var}(Y) \tag{19}$$
$$= \Sigma + \Delta_C \Delta_C^T \mathbb{E}[Y\pi^{(R|1)}\pi^{(B|1)} + (1 - Y)\pi^{(R|0)}\pi^{(B|0)}] + \bar{\Delta}\bar{\Delta}^T \pi^{(1)}\pi^{(0)} \tag{20}$$
$$= \Sigma + 2\pi_0(1 - 2\pi_0)\Delta_C \Delta_C^T + \frac{1}{4}\bar{\Delta}\bar{\Delta}^T. \tag{21}$$

Next, we compute $\text{Cov}(X, Y)$ as follows:

$$\text{Cov}(X, Y) = \mathbb{E}[\text{Cov}(X, Y|Y)] + \text{Cov}(\mathbb{E}[X|Y], \mathbb{E}[Y|Y]) \tag{22}$$
$$= \text{Cov}(\mathbb{E}[X|Y], Y) \tag{23}$$
$$= \text{Cov}(\mu^{(0)} + Y\bar{\Delta}, Y) \tag{24}$$
$$= \text{Cov}(Y\bar{\Delta}, Y) \tag{25}$$
$$= \bar{\Delta}\text{Var}(Y) \tag{26}$$
$$= \pi^{(1)}\pi^{(0)}\bar{\Delta} \tag{27}$$
$$= \frac{1}{4}\bar{\Delta}. \tag{28}$$

In order to write (21) and (28) only in terms of $\Delta_C$ and $\Delta_D$ and to see the effect of $\pi_0$, we show the following relationship between $\bar{\Delta}$, $\Delta_C$ and $\Delta_D$.

We introduce the following two minor lemmas that allow us to obtain clean expression for the optimal weights.

**Lemma A.3.** *Let $\bar{\Delta} := \mu^{(1)} - \mu^{(0)}$. Then*

$$\Delta_D - \bar{\Delta} = (1 - 4\pi_0)\Delta_C$$

*Proof.* We first note that

$$\mu^{(1)} = 2\pi_0(\mu^{(1,B)} - \mu^{(1,R)}) + \mu^{(1,R)} = 2\pi_0 \Delta_C + \mu^{(1,R)}.$$

Similarly,

$$\mu^{(0)} = 2\pi_0(\mu^{(0,R)} - \mu^{(0,B)}) + \mu^{(0,B)} = -2\pi_0 \Delta_C + \mu^{(0,B)}.$$

Combining these with the definitions of $\Delta_D$ and $\bar{\Delta}$, we get

$$
\begin{aligned}
\Delta_D - \bar{\Delta} &= \Delta_D - (\mu^{(1)} - \mu^{(0)}) \\
&= \mu^{(1,R)} - \mu^{(0,R)} - 2\pi_0 \Delta_C - \mu^{(1,R)} - 2\pi_0 \Delta_C + \mu^{(0,B)} \\
&= \mu^{(0,B)} - \mu^{(0,R)} - 4\pi_0 \Delta_C \\
&= (1 - 4\pi_0)\Delta_C.
\end{aligned}
$$

$\square$

From (21), (28) and Lemma A.3, we then obtain

$$
w_{\text{ERM}} = \frac{1}{4}\left(\Sigma + 2\pi_0(1 - 2\pi_0)\Delta_C \Delta_C^T + \frac{1}{4}\bar{\Delta}\bar{\Delta}^T\right)^{-1}\bar{\Delta}, \tag{29}
$$

where $\bar{\Delta} := \mu^{(1)} - \mu^{(0)} = \Delta_D - (1 - 4\pi_0)\Delta_C$, and

$$
b_{\text{ERM}} = \frac{1}{2} - \frac{1}{2}(w_{\text{ERM}})^T(\mu^{(0,R)} + \mu^{(1,B)}). \tag{30}
$$

Therefore,

$$
A^{(1,d)}(f_{\theta_{\text{ERM}}}) = \Phi\left(\frac{(w_{\text{ERM}})^T\left(\mu^{(1,d)} - \frac{1}{2}(\mu^{(0,R)} - \mu^{(1,B)})\right)}{\sqrt{(w_{\text{ERM}})^T \Sigma w_{\text{ERM}}}}\right), \tag{31}
$$

$$
A^{(0,d)}(f_{\theta_{\text{ERM}}}) = \Phi\left(\frac{-(w_{\text{ERM}})^T\left(\mu^{(0,d)} - \frac{1}{2}(\mu^{(0,R)} - \mu^{(1,B)})\right)}{\sqrt{(w_{\text{ERM}})^T \Sigma w_{\text{ERM}}}}\right). \tag{32}
$$

We can simplify the expressions in (31) and (32) by using the following relations:

$$
\mu^{(0,R)} - \frac{1}{2}(\mu^{(0,R)} - \mu^{(1,B)}) = \frac{1}{2}(\mu^{(0,R)} - \mu^{(1,B)}) = \frac{1}{2}(\mu^{(0,R)} - \mu^{(1,R)} + \mu^{(1,R)} - \mu^{(1,B)}) = -\frac{1}{2}(\Delta_C + \Delta_D),
$$

$$
\mu^{(0,B)} - \frac{1}{2}(\mu^{(0,R)} - \mu^{(1,B)}) = \frac{1}{2}(\mu^{(0,B)} - \mu^{(0,R)}) + \frac{1}{2}(\mu^{(0,B)} - \mu^{(1,B)}) = \frac{1}{2}(\Delta_C - \Delta_D),
$$

$$
\mu^{(1,B)} - \frac{1}{2}(\mu^{(0,R)} - \mu^{(1,B)}) = \frac{1}{2}(\mu^{(1,B)} - \mu^{(0,R)}) = \frac{1}{2}(\mu^{(1,B)} - \mu^{(1,R)} + \mu^{(1,R)} - \mu^{(0,R)}) = \frac{1}{2}(\Delta_C + \Delta_D),
$$

$$
\mu^{(1,R)} - \frac{1}{2}(\mu^{(0,R)} - \mu^{(1,B)}) = \frac{1}{2}(\mu^{(1,R)} - \mu^{(0,R)}) + \frac{1}{2}(\mu^{(1,R)} - \mu^{(1,B)}) = \frac{1}{2}(\Delta_D - \Delta_C).
$$

Plugging these into (31) and (32) for each group $(y, d) \in \{0, 1\} \times \{R, B\}$ yields

$$
A^{(0,R)}(f_{\theta_{\text{ERM}}}) = A^{(1,B)}(f_{\theta_{\text{ERM}}}) = \Phi\left(\frac{\frac{1}{2}(w_{\text{ERM}})^T(\Delta_C + \Delta_D)}{\sqrt{(w_{\text{ERM}})^T \Sigma w_{\text{ERM}}}}\right),
$$

$$
A^{(0,B)}(f_{\theta_{\text{ERM}}}) = A^{(1,R)}(f_{\theta_{\text{ERM}}}) = \Phi\left(\frac{\frac{1}{2}(w_{\text{ERM}})^T(\Delta_D - \Delta_C)}{\sqrt{(w_{\text{ERM}})^T \Sigma w_{\text{ERM}}}}\right).
$$

Thus,

$$
\text{WGA}(\theta_{\text{ERM}}) = \min\left\{\Phi\left(\frac{\frac{1}{2}(w_{\text{ERM}})^T(\Delta_C + \Delta_D)}{\sqrt{(w_{\text{ERM}})^T \Sigma w_{\text{ERM}}}}\right), \Phi\left(\frac{\frac{1}{2}(w_{\text{ERM}})^T(\Delta_D - \Delta_C)}{\sqrt{(w_{\text{ERM}})^T \Sigma w_{\text{ERM}}}}\right)\right\} \tag{33}
$$

In order to rewrite (29) to be able to simplify (33), we will use the following lemma.

**Lemma A.4.** *Let $A \in \mathbb{R}^{m \times m}$ be symmetric positive definite (SPD) and $u, v \in \mathbb{R}^m$. Then*

$$(A + vv^T + uu^T)^{-1}u = c_u \left( A^{-1}u - c_v A^{-1}v \right) \quad with \quad c_u := \frac{1}{1 + u^T B^{-1}u} \quad and \quad c_v := \frac{v^T A^{-1}u}{1 + v^T A^{-1}v}.$$

*Proof.* Let $B := A + vv^T$. Then

$$\begin{aligned}
(A + vv^T + uu^T)^{-1}u &= (B + uu^T)^{-1}u \\
&= \left( B^{-1} - \frac{B^{-1}uu^T B^{-1}}{1 + u^T B^{-1}u} \right) u \quad \text{(Sherman-Morrison formula)} \\
&= B^{-1}u - \frac{u^T B^{-1}u}{1 + u^T B^{-1}u} B^{-1}u \\
&= c_u B^{-1}u \\
&= c_u \left( A^{-1} - \frac{A^{-1}vv^T A^{-1}}{1 + v^T A^{-1}v} \right) u \quad \text{(Sherman-Morrison formula)}.
\end{aligned}$$

The assumption that $A$ is SPD guarantees that $A^{-1}$ exists, $B$ is SPD, and $c_u$ and $c_v$ are well-defined. $\qquad \square$

Applying Lemma A.4 to (29) with $A = \Sigma$, $u = \bar{\Delta}/2$ and $v = \sqrt{\beta}\Delta_C$, where $\beta := 2\pi_0(1 - 2\pi_0)$, yields

$$\begin{aligned}
w_{\text{ERM}} &= \gamma_{\text{ERM}} \left( \Sigma^{-1}\bar{\Delta} - \frac{\beta \Delta_C^T \Sigma^{-1}\bar{\Delta}}{1 + \beta \Delta_C^T \Sigma^{-1}\Delta_C} \Sigma^{-1}\Delta_C \right) \\
&= \gamma_{\text{ERM}} \left( \Sigma^{-1}\Delta_D - \frac{\delta + \beta \Delta_C^T \Sigma^{-1}\Delta_D}{1 + \beta \Delta_C^T \Sigma^{-1}\Delta_C} \Sigma^{-1}\Delta_C \right) \quad \text{(substituting } \bar{\Delta} = \Delta_D - \delta\Delta_C) \\
&= \gamma_{\text{ERM}} \left( \Sigma^{-1}\Delta_D - c_{\pi_0} \Sigma^{-1}\Delta_C \right)
\end{aligned}$$

with

$$\gamma_{\text{ERM}} := \frac{1}{4 + \bar{\Delta}^T A_{\text{ERM}}^{-1}\bar{\Delta}} \quad and \quad c_{\pi_0} := \frac{\delta + \beta \Delta_C^T \Sigma^{-1}\Delta_D}{1 + \beta \Delta_C^T \Sigma^{-1}\Delta_C}.$$

Let $\|v\| := \sqrt{v^T \Sigma^{-1}v}$ be the norm induced by the $\Sigma^{-1}$–inner product. Then

$$\text{WGA}(\theta_{\text{ERM}}) = \min \left\{ \Phi \left( \frac{(1 - c_{\pi_0})\Delta_C^T \Sigma^{-1}\Delta_D + \|\Delta_D\|^2 - c_{\pi_0}\|\Delta_C\|^2}{2\|\Delta_D - c_{\pi_0}\Delta_C\|} \right), \right.$$
$$\left. \Phi \left( \frac{-(c_{\pi_0} + 1)\Delta_C^T \Sigma^{-1}\Delta_D + \|\Delta_D\|^2 + c_{\pi_0}\|\Delta_C\|^2}{2\|\Delta_D - c_{\pi_0}\Delta_C\|} \right) \right\}$$

Under Assumption 3.6, $c_{\pi_0} = \tilde{c}_{\pi_0} := (1 - 4\pi_0)/(1 + 2\pi_0(1 - 2\pi_0)\|\Delta_C\|^2)$ and

$$\text{WGA}(\theta_{\text{ERM}}) = \min \left\{ \Phi \left( \frac{\|\Delta_D\|^2 - \tilde{c}_{\pi_0}\|\Delta_C\|^2}{2\sqrt{\|\Delta_D\|^2 + \tilde{c}_{\pi_0}^2\|\Delta_C\|^2}} \right), \Phi \left( \frac{\|\Delta_D\|^2 + \tilde{c}_{\pi_0}\|\Delta_C\|^2}{2\sqrt{\|\Delta_D\|^2 + \tilde{c}_{\pi_0}^2\|\Delta_C\|^2}} \right) \right\}, \qquad (34)$$

where the first term is the accuracy of the minority groups and the second is that of the majority groups. In order to compare the WGA of ERM with the WGA of DS, we first show that under Assumption 3.6 the WGA of ERM is given by the majority accuracy term in (34). Since $\tilde{c}_{\pi_0} \geq 0$ for $\pi_0 \leq 1/4$, we have that

$$\frac{\|\Delta_D\|^2 - \tilde{c}_{\pi_0}\|\Delta_C\|^2}{2\sqrt{\|\Delta_D\|^2 + \tilde{c}_{\pi_0}^2\|\Delta_C\|^2}} \leq \frac{\|\Delta_D\|^2 + \tilde{c}_{\pi_0}\|\Delta_C\|^2}{2\sqrt{\|\Delta_D\|^2 + \tilde{c}_{\pi_0}^2\|\Delta_C\|^2}} \quad \Leftrightarrow \quad \tilde{c}_{\pi_0}\|\Delta_C\|^2 \geq 0,$$

which is satisfied for all $\pi_0 \leq 1/4$ with equality at $\pi_0 = 1/4$. Since $\Phi$ is increasing, we get

$$\text{WGA}(\theta_{\text{ERM}}) = \Phi \left( \frac{\|\Delta_D\|^2 - \tilde{c}_{\pi_0}\|\Delta_C\|^2}{2\sqrt{\|\Delta_D\|^2 + \tilde{c}_{\pi_0}^2\|\Delta_C\|^2}} \right). \qquad (35)$$

$\square$

**Lemma A.5.** *Learning the model in* (3) *after downsampling according to noisy domain labels using the noisy minority prior* $\pi_0^{(p)} := (1-p)\pi_0 + p(1/2 - \pi_0)$ *for* $p \in [0, 1/2]$ *is equivalent to learning the model with clean domain labels (no noise) and using the minority prior*

$$\pi_{DS}^{(p)} := \frac{(1-p)\pi_0}{4\pi_0^{(p)}} + \frac{p\pi_0}{4(1/2 - \pi_0^{(p)})}. \tag{36}$$

*Proof.* We note that the model learned after downsampling is agnostic to domain labels, so only the true proportion of each group, not the noisy proportion, determines the model weights. We derive the equivalent *clean* prior. We do so by examining how true minority samples are affected by DS on the data with noisy domain labels. When DS is performed on the data with domain label noise, the true minority samples that are kept can be categorized as (i) those that are still minority samples in the noisy data and (ii) a proportion of those that have become majority samples in the noisy data.

The first type of samples appear with probability

$$(1-p)\pi_0, \tag{37}$$

i.e., the proportion of true minority samples whose domain was not flipped. The second type of samples are kept with probability

$$p\pi_0 \left( \frac{\pi_0^{(p)}}{1/2 - \pi_0^{(p)}} \right), \tag{38}$$

where the factor dependent on $\pi_0^{(p)}$ is the factor by which the size of the noisy majority groups will be reduced to be the same size as the noisy minority groups.

Therefore, the unnormalized true minority prior can be written as

$$(1-p)\pi_0 + p\pi_0 \left( \frac{\pi_0^{(p)}}{1/2 - \pi_0^{(p)}} \right). \tag{39}$$

We can repeat the same analysis for the majority groups to obtain the unnormalized true majority prior as

$$p(1/2 - \pi_0) + (1-p)(1/2 - \pi_0) \left( \frac{\pi_0^{(p)}}{1/2 - \pi_0^{(p)}} \right). \tag{40}$$

In order for the true minority and true majority priors to sum to one over the four groups, we divide by the normalization factor $4\pi_0^{(p)}$, so our final minority prior is given by

$$\frac{(1-p)\pi_0 + p\pi_0 \left( \frac{\pi_0^{(p)}}{1/2 - \pi_0^{(p)}} \right)}{4\pi_0^{(p)}} = \frac{(1-p)\pi_0}{4\pi_0^{(p)}} + \frac{p\pi_0}{4(1/2 - \pi_0^{(p)})}. \tag{41}$$

$\square$

DS is usually a special case of ERM with $\pi_0 = 1/4$. However, since DS uses domain labels and therefore is not agnostic to noise, we need to use the prior derived in Lemma A.5 to be able to analyze the effect of the noise $p$ while still using the clean data parameters. Note that $\pi_{DS}^{(p)}$ defined in (36) decreases from $1/4$ to $\pi_0$ as the noise $p$ increases from 0 to $1/2$. Since we can interpolate between $1/4$ to $\pi_0$ using $p$, we can therefore substitute $\pi_0$ in (12) with $\pi_{DS}^{(p)}$ and then examine the resulting expression as a function of $p$ for any $\pi_0$. Using

the WGA of ERM from Lemma A.2, we take the following derivative:

$$\frac{\partial}{\partial p} \frac{\|\Delta_D\|^2 - \tilde{c}_{\pi_{\mathrm{DS}}^{(p)}}\|\Delta_C\|^2}{2\sqrt{\|\Delta_D\|^2 + \tilde{c}_{\pi_{\mathrm{DS}}^{(p)}}^2\|\Delta_C\|^2}} = \frac{-\|\Delta_D\|^2\|\Delta_C\|^2\left(\tilde{c}_{\pi_{\mathrm{DS}}^{(p)}} + 1\right)}{2\left(\|\Delta_D\|^2 + \tilde{c}_{\pi_{\mathrm{DS}}^{(p)}}^2\|\Delta_C\|^2\right)^{3/2}} \times \frac{-2\left(16\left(\pi_{\mathrm{DS}}^{(p)}\right)^2 - 8\pi_{\mathrm{DS}}^{(p)} + 3\right)\|\Delta_C\|}{\left(1 + 2\pi_{\mathrm{DS}}^{(p)}\left(1 - 2\pi_{\mathrm{DS}}^{(p)}\right)\|\Delta_C\|^2\right)^2}$$

$$\times \frac{\pi_0(4\pi_0 - 1)(2\pi_0 - 1)(2p - 1)}{2\left(\pi_0(2 - 4p) + p\right)^2\left(\pi_0(4p - 2) - p + 1\right)^2},$$

which is strictly negative for $p < 1/2$ and $\pi_0 < 1/4$. Therefore, for any $\pi_0 < 1/4$, the WGA of DS is strictly decreasing in $p$ and recovers the WGA of ERM when $p = 1/2$ or when $\pi_0 = 1/4$. Thus, for $p \leq 1/2$ and $\pi_0 \leq 1/4$,

$$\mathrm{WGA}(\theta_{\mathrm{ERM}}) \leq \mathrm{WGA}(\theta_{\mathrm{DS}}^{(p)}) = \Phi\left(\frac{\|\Delta_D\|^2 - \tilde{c}_{\pi_{\mathrm{DS}}^{(p)}}\|\Delta_C\|^2}{2\left(\|\Delta_D\| + \tilde{c}_{\pi_{\mathrm{DS}}^{(p)}}\|\Delta_C\|\right)}\right), \tag{42}$$

with equality when $p = 1/2$ or $\pi_0 = 1/4$. Additionally, by Proposition 3.1,

$$\mathrm{WGA}(\theta_{\mathrm{UW}}^{(p)}) = \mathrm{WGA}(\theta_{\mathrm{DS}}^{(p)}) = \Phi\left(\frac{\|\Delta_D\|^2 - \tilde{c}_{\pi_{\mathrm{DS}}^{(p)}}\|\Delta_C\|^2}{2\left(\|\Delta_D\| + \tilde{c}_{\pi_{\mathrm{DS}}^{(p)}}\|\Delta_C\|\right)}\right). \tag{43}$$

## B  Datasets

Table 2: Dataset splits

| Dataset | Group, $g$ | | Data Quantity | | |
| --- | --- | --- | --- | --- | --- |
| | Class, $y$ | Domain, $d$ | Train | Val | Test |
| CelebA | non-blond | female | 71629 | 8535 | 9767 |
| | non-blond | male | 66874 | 8276 | 7535 |
| | blond | female | 22880 | 2874 | 2480 |
| | blond | male | 1387 | 182 | 180 |
| Waterbirds | landbird | land | 3506 | 462 | 2255 |
| | landbird | water | 185 | 462 | 2255 |
| | waterbird | land | 55 | 137 | 642 |
| | waterbird | water | 1049 | 138 | 642 |
| CMNIST | 0-4 | green | 1527 | 747 | 786 |
| | 0-4 | red | 13804 | 6864 | 6868 |
| | 5-9 | green | 13271 | 6654 | 6639 |
| | 5-9 | red | 1398 | 735 | 707 |
| MultiNLI | contradiction | no negation | 57498 | 22814 | 34597 |
| | contradiction | negation | 11158 | 4634 | 6655 |
| | entailment | no negation | 67376 | 26949 | 40496 |
| | entailment | negation | 1521 | 613 | 886 |
| | neither | no negation | 66630 | 26655 | 39930 |
| | neither | negation | 1992 | 797 | 1148 |

## C  Experimental Design

For the image datasets, we use a Resnet50 model pre-trained on ImageNet, imported from `torchvision` as the upstream model. For the text datasets, we use a BERT model pre-trained on Wikipedia, imported from the `transformers` package as the upstream model. In our experiments, we assume access to a validation set with clean domain annotations, which we use to tune the hyperparameters. For all methods, we tune the inverse of $\lambda$, where $\lambda$ is the regularization strength, over 20 (equally-spaced on a log scale) values ranging from

$1e - 4$ to 1. RAD-UW has more hyperparameters to tune, which we discuss in detail below for each dataset. For all the methods we tested, for each noise level, we perform ten rounds of hyperparameter tuning over ten seeds of adding domain annotation noise. We then fix the most commonly chosen value across the ten rounds. With the fixed hyperparameters, we rerun the algorithms over 10 random seeds of noise and report the mean and standard deviation across the ten seeds. RAD-UW uses a regularized linear model implemented with `pytorch` for the pseudo-annotation of domain labels (henceforth referred to as the *pseudo-annotation model*) and uses `LogisticRegression` model imported from `sklearn.linear_model` as the retraining model with upweighting. The pseudo-annotation model uses a weight decay of $1e - 3$ with the `AdamW` optimizer from `pytorch`. For all datasets except Waterbirds, the pseudo-annotation model is trained for 6 epochs. Waterbirds is trained for 60 epochs. For all datasets except CMNIST, we tune the inverse of $\lambda_{\text{ID}}$ for the pseudo-annotation model over 20 (equally-spaced on a log scale) values ranging from $1e - 7$ to $1e - 3$. For CMNIST, we tune the inverse of $\lambda_{\text{ID}}$ 20 (equally-spaced on a log scale) values ranging from $1e - 1$ to $1e2$. For the retraining model, we tune the inverse of $\lambda$ over 20 (equally-spaced on a log scale) values ranging from $1e - 4$ to 1.

## C.1   Waterbirds

For the upstream ResNet50 model, we use a constant learning rate of $1e - 3$, momentum of 0.9, and weight decay of $1e-3$. We train the upstream model for 100 epochs. We leverage random crops and random horizontal flips as data-augmentation during the training. For RAD-UW, we tune $\lambda$ for both the pseudo-annotation model and the retraining model. We tune the upweighting factor, $c$, for the retraining model over 5 equally spaced values ranging from 5 to 20. The pseudo-annotation model uses a constant learning rate of $1e - 5$.

## C.2   CelebA

For the upstream ResNet50 model, we use a constant learning rate of $1e - 3$, momentum of 0.9 and weight decay of $1e - 4$. We train the upstream model for 50 epochs while using random crops and random horizontal flips for data augmentation. We tune the upweighting factor, $c$, for the retraining model over 5 equally spaced values ranging from 20 to 40. The pseudo-annotation model is trained with an initial learning rate of $1e - 5$ with `CosineAnnealingLR` learning rate scheduler from `pytorch`.

## C.3   CMNIST

For the ResNet50 model, we use a constant learning rate of $1e - 3$, momentum of 0.9 and weight decay of $1e - 3$. We train the model for 10 epochs without any data augmentation. We tune the upweighting factor, $c$, over 5 equally spaced values ranging from 20 to 40. The pseudo-annotation model is trained with an initial learning rate of $1e - 5$ with `CosineAnnealingLR` learning rate scheduler from `pytorch`. The spurious correlations in CMNIST is between the digit being between less than 5 and the color of the digit. We note that this simple correlation is quite easy to learn, so the $\ell_1$ penalty needed in quite small.

## C.4   MultiNLI

We train the BERT model using code adapted from (Izmailov et al., 2022). We train the model for 10 epochs with an initial learning rate of $1e - 4$ and a weight decay of $1e - 4$. We use the `linear` learning rate scheduler imported from the `transformers` library. The upweight factor is tuned over 5 equally spaced values ranging from 4 to 10. The pseudo-annotation model is trained with an initial learning rate of $1e - 4$ with `CosineAnnealingLR` learning rate scheduler from `pytorch`.

## C.5   Civil Comments

Similar to MultiNLI, We train the BERT model using code adapted from (Izmailov et al., 2022). We train the model for 10 epochs with an initial learning rate of $1e - 5$ and a weight decay of $1e - 4$. We use the `linear` learning rate scheduler imported from the `transformers` library. The upweight factor is tuned over 5 equally spaced values ranging from 4 to 10. The pseudo-annotation model is trained with an initial learning rate of $1e - 4$ with `CosineAnnealingLR` learning rate scheduler from `pytorch`.

Table 3: M-SELF Hyperparameters

| Dataset | fine-tuning steps | learning rate range | num points range |
|---|---|---|---|
| CelebA | 500 | [1e-6, 1e-5, 1e-4] | [2, 20, 100] |
| Waterbirds | 500 | [1e-4, 1e-3, 1e-2] | [20, 100, 500] |
| CMNIST | 500 | [1e-5, 1e-4, 1e-3] | [100, 500, 700] |
| Civilcomments | 200 | [1e-6, 1e-5, 1e-4] | [20, 100, 500] |

Table 4: CelebA group downsampling WGA

| Method | Label Noise | | | | |
|---|---|---|---|---|---|
| | 0% | 5% | 10% | 15% | 20% |
| Averaged | 87.33±0.46 | 83.78±1.33 | 79.22±2.14 | 78.67±3.78 | 77.33±3.03 |
| Single | 84.11±3.25 | 81.56±3.05 | 76.44±3.82 | 78.67±4.67 | 76.89±3.06 |

## D  M-SELF Implementation

We implemented misclassification-SELF using code adapted from LaBonte et al. (2023) so that it would be compatible with our setup, where we use pre-generated embeddings from the base models. We fix the `finetuning steps`, which is the number of steps of fine-tuning we perform once we construct the class-balanced error set. We tune the learning rate and the number of points that are selected for class balancing. The hyperparameter values and ranges used are given in Table 3

## E  Model Averaging

DFR (Kirichenko et al., 2023) emphasizes the importance of model averaging in their setting as seeing different downsampled retraining sets can dramatically change the fair classifier. To examine these effects we train GDS 10 times and average the weights over those 10 runs. We report the mean and standard deviation of the worst-group accuracy of this averaged model over 10 independent runs.

We see in Tables 4 to 7 that for each dataset, averaging helps to reduce variance and provides modest increases in performance for both CelebA and CMNIST. This same model averaging could be performed both for SELF and RAD, but would have no effect in GUW as the weighting set is fixed as is the upweighting factor.

## F  Additional Tables

Methods which require domain annotations (DA) are denoted with a "Y" in the appropriate column, while those that are agnostic to DA are denoted as "N". We collate the results of similar methods and separate those for different approaches in our tables by horizontal lines. Finally, results for methods designed to annotate domains before using data augmentations, namely RAD-UW and SELF (LaBonte et al., 2023), are collected at the bottom of each table.

## G  Robustness to Noisy Embeddings

We note in Table 13 and Table 14 that M-SELF has significantly worse performance than RAD-UW when using noisy embeddings, likely because M-SELF additionally reuses the final classification layer of the base

Table 5: CMNIST group downsampling WGA

| Method | Label Noise | | | | |
|---|---|---|---|---|---|
| | 0% | 5% | 10% | 15% | 20% |
| Averaged | 96.12±0.21 | 95.08±0.25 | 94.62±0.41 | 94.42±0.54 | 93.75±0.59 |
| Single | 95.76±0.33 | 94.42±0.55 | 94.31±0.34 | 94.26±0.64 | 93.41±0.73 |

Table 6: Waterbirds group downsampling WGA

| Method | Label Noise | | | | |
|---|---|---|---|---|---|
| | 0% | 5% | 10% | 15% | 20% |
| Averaged | 91.78±0.68 | 91.81±0.81 | 92.06±0.62 | 91.78±0.68 | 91.84±0.67 |
| Single | 91.59±0.61 | 91.40±0.51 | 91.34±0.65 | 91.78±0.48 | 91.00±0.78 |

Table 7: CivilComments group downsampling WGA

| Method | Label Noise | | | | |
|---|---|---|---|---|---|
| | 0% | 5% | 10% | 15% | 20% |
| Averaged | 80.45±0.43 | 80.27±0.35 | 80.73±0.57 | 80.64±0.64 | 80.57±0.86 |
| Single | 80.47±0.33 | 80.11±0.47 | 80.64±0.61 | 80.53±0.57 | 80.45±1.10 |

model. Previous work (Iscen et al., 2019; 2022) has shown that most of the drop in performance when training deep models with noisy data comes from a poor classification head.

Table 8: **CMNIST WGA under domain label noise.** CMNIST has a relatively small spurious correlation and as such even LLR performs quite well. Additionally, CMNIST is already class balanced so class downsampling and upweighting have no effect. (LaBonte et al., 2023) do not provide WGA for CMNIST; RAD-UW matches the performance of group-dependent methods at 10% and surpasses it beyond that.

| Method | DA | Domain Annotation Noise (%) | | | | |
| --- | --- | --- | --- | --- | --- | --- |
| | | 0 | 5 | 10 | 15 | 20 |
| Group Downsample | Y | $\mathbf{95.73 \pm 0.58}$ | $\mathbf{94.77 \pm 0.46}$ | $\mathbf{94.26 \pm 0.33}$ | $93.90 \pm 0.58$ | $93.15 \pm 0.41$ |
| Group Upweight | Y | $95.19 \pm 0$ | $94.60 \pm 0.22$ | $94.03 \pm 0.22$ | $93.54 \pm 0.31$ | $93.12 \pm 0.28$ |
| Class Downsample | N | $91.47 \pm 0.13$ | $91.47 \pm 0.13$ | $91.47 \pm 0.13$ | $91.47 \pm 0.13$ | $91.47 \pm 0.13$ |
| Class Upweight | N | $91.94 \pm 0.09$ | $91.94 \pm 0.09$ | $91.94 \pm 0.09$ | $91.94 \pm 0.09$ | $91.94 \pm 0.09$ |
| LLR | N | $91.81 \pm 0.08$ | $91.81 \pm 0.08$ | $91.81 \pm 0.08$ | $91.81 \pm 0.08$ | $91.81 \pm 0.08$ |
| RAD-UW | N | $94.06 \pm 0.15$ | $94.06 \pm 0.15$ | $94.06 \pm 0.15$ | $\mathbf{94.06 \pm 0.15}$ | $\mathbf{94.06 \pm 0.15}$ |
| M-SELF (Our Imp.) | N | $92.04 \pm 0.20$ | $92.04 \pm 0.20$ | $92.04 \pm 0.20$ | $92.04 \pm 0.20$ | $92.04 \pm 0.20$ |

Table 9: **CelebA WGA under domain label noise.** RAD-UW outperforms domain annotation-dependent methods at only 5% noise and existing misclassification-based domain annotation-free baselines at every noise level.

| Method | DA | Domain Annotation Noise (%) | | | | |
| --- | --- | --- | --- | --- | --- | --- |
| | | 0 | 5 | 10 | 15 | 20 |
| Group Downsample | Y | $86.05 \pm 0.84$ | $\mathbf{84.05 \pm 1.83}$ | $83.00 \pm 1.17$ | $81.67 \pm 1.89$ | $78.83 \pm 1.98$ |
| Group Upweight | Y | $\mathbf{86.11 \pm 0}$ | $82.55 \pm 0.67$ | $82.39 \pm 0.43$ | $81.72 \pm 0.52$ | $78.72 \pm 0.96$ |
| Class Downsample | N | $74.5 \pm 0.94$ | $74.5 \pm 0.94$ | $74.5 \pm 0.94$ | $74.5 \pm 0.94$ | $74.5 \pm 0.94$ |
| Class Upweight | N | $73.89 \pm 0.00$ | $73.89 \pm 0.00$ | $73.89 \pm 0.00$ | $73.89 \pm 0.00$ | $73.89 \pm 0.00$ |
| LLR | N | $44.28 \pm 0.5$ | $44.28 \pm 0.5$ | $44.28 \pm 0.5$ | $44.28 \pm 0.5$ | $44.28 \pm 0.5$ |
| RAD-UW | N | $83.51 \pm 0.02$ | $83.51 \pm 0.02$ | $\mathbf{83.51 \pm 0.02}$ | $\mathbf{83.51 \pm 0.02}$ | $\mathbf{83.51 \pm 0.02}$ |
| M-SELF (Our Imp.) | N | $83.48 \pm 0.02$ | $83.48 \pm 0.02$ | $83.48 \pm 0.02$ | $83.48 \pm 0.02$ | $83.48 \pm 0.02$ |
| M-SELF (LaBonte et al.) | N | $83.0 \pm 6.1$ | $83.0 \pm 6.1$ | $83.0 \pm 6.1$ | $83.0 \pm 6.1$ | $83.0 \pm 6.1$ |

Table 10: **Waterbirds WGA under domain label noise.** The validation (retraining) split for Waterbirds is domain balanced already, so class and group balancing perform equivalently. All domain annotation-free methods show improvement over baselines.

| Method | DA | Domain Annotation Noise (%) | | | | |
| --- | --- | --- | --- | --- | --- | --- |
| | | 0 | 5 | 10 | 15 | 20 |
| Group Downsample | Y | $92.14 \pm 0.99$ | $91.73 \pm 0.44$ | $92.07 \pm 0.27$ | $92.19 \pm 0.73$ | $92.44 \pm 0.65$ |
| Group Upweight | Y | $90.92 \pm 0.07$ | $90.98 \pm 0.13$ | $90.95 \pm 0.15$ | $90.86 \pm 0.16$ | $90.89 \pm 0.25$ |
| Class Downsample | N | $91.44 \pm 0.31$ | $91.44 \pm 0.31$ | $91.44 \pm 0.31$ | $91.44 \pm 0.31$ | $91.44 \pm 0.31$ |
| Class Upweight | N | $91.40 \pm 0.06$ | $91.40 \pm 0.06$ | $91.40 \pm 0.06$ | $91.40 \pm 0.06$ | $91.40 \pm 0.06$ |
| LLR | N | $87.35 \pm 0.06$ | $87.35 \pm 0.06$ | $87.35 \pm 0.06$ | $87.35 \pm 0.06$ | $87.35 \pm 0.06$ |
| RAD-UW | N | $\mathbf{92.52 \pm 0}$ | $\mathbf{92.52 \pm 0}$ | $\mathbf{92.52 \pm 0}$ | $\mathbf{92.52 \pm 0}$ | $\mathbf{92.52 \pm 0}$ |
| M-SELF (Our Imp.) | N | $\mathbf{92.83 \pm 0.49}$ | $\mathbf{92.83 \pm 0.49}$ | $\mathbf{92.83 \pm 0.49}$ | $\mathbf{92.83 \pm 0.49}$ | $\mathbf{92.83 \pm 0.49}$ |
| M-SELF (LaBonte et al.) | N | $\mathbf{92.6 \pm 0.8}$ | $\mathbf{92.6 \pm 0.8}$ | $\mathbf{92.6 \pm 0.8}$ | $\mathbf{92.6 \pm 0.8}$ | $\mathbf{92.6 \pm 0.8}$ |

Table 11: **MultiNLI WGA under domain label noise.** SELF's reported result performs strongly on this dataset, perhaps due to the structure of the spurious correlation. We were unable to replicate the results of LaBonte et al. (2023) with our implementation. Regardless, both SELF and RAD-UW outperform group-dependent methods at label noise above 5%.

| Method | DA | Domain Annotation Noise (%) | | | | |
|---|---|---|---|---|---|---|
| | | 0 | 5 | 10 | 15 | 20 |
| Group Downsample | Y | $73.19 \pm 1.16$ | $67.96 \pm 1.14$ | $65.94 \pm 0.39$ | $64.93 \pm 0.24$ | $65.31 \pm 0.13$ |
| Group Upweight | Y | $\mathbf{74.31 \pm 0.07}$ | $69.21 \pm 0.46$ | $66.22 \pm 0.24$ | $65.26 \pm 0.24$ | $65.87 \pm 0.19$ |
| Class Downsample | N | $65.50 \pm 0.07$ | $65.50 \pm 0.07$ | $65.50 \pm 0.07$ | $65.50 \pm 0.07$ | $65.50 \pm 0.07$ |
| Class Upweight | N | $65.05 \pm 0.15$ | $65.05 \pm 0.15$ | $65.05 \pm 0.15$ | $65.05 \pm 0.15$ | $65.05 \pm 0.15$ |
| LLR | N | $65.36 \pm 0.35$ | $65.47 \pm 0.31$ | $65.47 \pm 0.31$ | $65.47 \pm 0.31$ | $65.47 \pm 0.31$ |
| RAD-UW | N | $68.53 \pm 0.14$ | $68.53 \pm 0.14$ | $68.53 \pm 0.14$ | $68.53 \pm 0.14$ | $68.53 \pm 0.14$ |
| M-SELF (Our Imp.) | N | $64.02 \pm 0$ | $64.02 \pm 0$ | $64.02 \pm 0$ | $64.02 \pm 0$ | $64.02 \pm 0$ |
| M-SELF (LaBonte et al.) | N | $72.2 \pm 2.2$ | $\mathbf{72.2 \pm 2.2}$ | $\mathbf{72.2 \pm 2.2}$ | $\mathbf{72.2 \pm 2.2}$ | $\mathbf{72.2 \pm 2.2}$ |

Table 12: **CivilComments WGA under domain label noise.** RAD-UW dramatically outperforms SELF which is unable to learn with the severe class imbalance present in CivilComments.

| Method | DA | Domain Annotation Noise (%) | | | | |
|---|---|---|---|---|---|---|
| | | 0 | 5 | 10 | 15 | 20 |
| Group Downsample | Y | $80.66 \pm 0.21$ | $80.42 \pm 0.24$ | $81.14 \pm 0.09$ | $81.07 \pm 0.15$ | $81.25 \pm 0.17$ |
| Group Upweight | Y | $81.08 \pm 0.01$ | $80.96 \pm 0.02$ | $80.87 \pm 0.01$ | $80.78 \pm 0.03$ | $80.73 \pm 0.02$ |
| Class Downsample | N | $81.56 \pm 0.12$ | $81.56 \pm 0.12$ | $81.56 \pm 0.12$ | $81.56 \pm 0.12$ | $81.56 \pm 0.12$ |
| Class Upweight | N | $81.50 \pm 0.03$ | $81.50 \pm 0.03$ | $81.50 \pm 0.03$ | $81.50 \pm 0.03$ | $81.50 \pm 0.03$ |
| LLR | N | $58.59 \pm 0.01$ | $58.59 \pm 0.01$ | $58.59 \pm 0.01$ | $58.59 \pm 0.01$ | $58.59 \pm 0.01$ |
| RAD-UW | N | $\mathbf{81.57 \pm 0.03}$ | $\mathbf{81.57 \pm 0.03}$ | $\mathbf{81.57 \pm 0.03}$ | $\mathbf{81.57 \pm 0.03}$ | $\mathbf{81.57 \pm 0.03}$ |
| M-SELF (Our Imp.) | N | $60.61 \pm 0.04$ | $60.61 \pm 0.04$ | $60.61 \pm 0.04$ | $60.61 \pm 0.04$ | $60.61 \pm 0.04$ |
| M-SELF (LaBonte et al.) | N | $62.7 \pm 4.6$ | $62.7 \pm 4.6$ | $62.7 \pm 4.6$ | $62.7 \pm 4.6$ | $62.7 \pm 4.6$ |

Table 13: **CelebA WGA (std. dev.)** using embeddings from a noisy base model. We see that RAD-UW is significantly more robust to than M-SELF as M-SELF reuses the base model's noisy classifier.

| | Base Model Label Noise | |
|---|---|---|
| Method | 0 % | 20% |
| RAD-UW | $83.51 \pm 0.02$ | $80.55 \pm 0.02$ |
| M-SELF | $83.48 \pm 0.02$ | $31.86 \pm 2.32$ |

Table 14: **Waterbirds WGA (std. dev.)** using embeddings from a noisy base model. Again RAD-UW is more robust than M-SELF.

| | Base Model Label Noise | |
|---|---|---|
| Method | 0% | 20% |
| RAD-UW | $92.52 \pm 0$ | $86.12 \pm 0.05$ |
| M-SELF | $92.83 \pm 0.49$ | $67.43 \pm 4.48$ |

