# OpenReview forum: "For Robust Worst-Group Accuracy, Ignore Group Annotations"
_TMLR — Accepted by TMLR_

### Review · Reviewer_ct71 · 2024-07-09

**Summary Of Contributions:**

This paper deals with the challege of optimizing worst-group accuracy under the setting of training with imbalanced data where the domain labels are noisy. The authors first present a theoretical analysis arguing that the two common methods in the literature of optimizing WGA have the same theoretical guarantee, which degrades as the domain labels get noisier. They further propose to use an explicitly regularized model that learns spuriously correlated features to label minority data to enhance the common UW method. The experimental results show that the proposed method achieves competitive results.

**Audience:**

Yes

**Broader Impact Concerns:**

I have no concerns about the ethical implications of the work.

**Claims And Evidence:**

Yes

**Requested Changes:**

Please see the weaknesses stated above.

**Strengths And Weaknesses:**

Strengths:
- The target problem is clearly stated and motivated.

- The related works section covers the relevant approaches.

- The theoretical analysis provides a deeper understanding of the two common methods in this field.

- The methodology is clearly explained.

Weaknesses:

- In the abstract, the authors state: “Our results show that RAD is competitive with other recently proposed domain annotation-free techniques. Most importantly, RAD outperforms state-of-the-art annotation-reliant methods even with only 5% noise in the training data for several publicly available datasets.” In my understanding, these mean that in terms of performance, recent domain annotation-free methods >= this paper’s method > SOTA domain annotation-reliant methods. Since the annotation-free methods do not require domain labels and thus cost less than annotation-reliant methods (meanwhile, perform better), I cannot see why the comparison with domain-reliant methods matters.

- I think that conceptually, there are only two differences between the proposed RAD-UW and M-SELF proposed by LaBonte et al (2023): RAD-UW uses explicit l1 regularization for domain labelling and adopts UW for WGA optimizing, whereas M-SELF uses implicit early-stopping regularization and adopts DS. As l1 regularization is a basic concept in ML and UW is a common method in optimizing WGA, I think the novelty of the proposed method is quite limited. The experimental results showing that the proposed method failed to outperform existing methods in many settings also limit its significance.

---

> ### Author Response · Authors · 2024-09-02
>
> ### Rebuttal
> We thank the reviewer for their comments and appreciate that they find our work well-written and motivated. We would like to address each of their concerns, denoting weaknesses as W.
>
>  - **(W1)**	We emphasize that this ordering only holds with noise in the finetuning set, as we mention. In fact at 0% noise, domain-reliant methods outperform both RAD and SELF on nearly every dataset. This is to be expected as RAD and SELF are essentially trying to predict domain labels whereas domain-reliant methods have access to the ground truth (at 0% noise). A central point of the paper is to emphasize that domain-reliant methods struggle when noise is present, and the ordering that is pointed out is a consequence of that.
>  - **(W2)**	We would like to politely push back on the fact that RAD has limited novelty in the face of M-SELF, as we believe that RAD cleverly combines existing methodologies to create a method with strong performance and surprisingly increased robustness.
> We would like to expound a bit more of the differences between M-SELF and RAD and how they affect performance.
>     - First, the classification head used by M-SELF is the pretrained classifier which was jointly trained with the embeddings. This is a reasonable baseline, but does not exploit recent insights into the structure of spuriously correlated classification, e.g. Kirichenko, et al. RAD’s utilization of a strong L1 penalty cleverly adapts the findings of Kirichenko, et al. to train a classifier which is strongly biased towards spurious features. This allows for minority samples to be more easily identified, but has some hidden robustness as well.
>     - In our exploration of these methods, we found that when using embeddings which were trained on noisy data, both methods take some performance hit, but RAD is dramatically more robust than SELF. This is because the embeddings themselves are fairly robust to label noise, but the final classification head is more negatively affected (see, for example, Iscen, et al. (2022)). Thus the reuse of this classifier negatively impacts the robustness of SELF to an imperfect base model. We also performed such ablation studies ourselves and have included them as tables below comparing the two methods using embeddings from a clean and noisy base model for two distinct datasets.
>     - Finally, utilizing upweighting as opposed to downsampling greatly reduces the variance of RAD as opposed to SELF and is particularly helpful when data is limited.
>
> We hope that these additional details help to clarify some of the confusion and emphasize the strength of RAD as compared to existing methods.
>
>
> ### References
> A. Iscen, J. Valmadre, A. Arnab, and C. Schmid. Learning with neighbor consistency for noisy labels. In Proceedings of the IEEE/CVF Conference on Computer Vision and Pattern Recognition, pages 4672–4681, 2022
>
> ### Tables
> **CelebA**, embeddings trained with 20% SLN
> | Method | Clean Embeddings WGA | Noisy Embeddings WGA |
> |--------|----------------------|----------------------|
> | RAD    | 83.51 ± 0.02         | 80.55 ± 0.02         |
> | SELF   | 83.48 ± 0.02         | 31.86 ± 2.32         |
>
> **Waterbirds**, embeddings trained with 20% SLN
> | Method | Clean Embeddings WGA | Noisy Embeddings WGA |
> |--------|----------------------|----------------------|
> | RAD    | 92.52 ± 0            | 86.12 ± 0.05         |
> | SELF   | 92.83 ± 0.49         | 67.43 ± 4.48         |

---

### Review · Reviewer_7RCn · 2024-07-17

**Summary Of Contributions:**

The paper proposes a novel approach for worst-group accuracy optimization, where a classifiers a trained to overcome spurious correlations by focussing on achieving sufficient performance for all available groups. Groups are determined by an extra set of labels, although several approaches exist which achieve the same goal without using them.

The first contribution of the paper are a theortical analysis of downsampling and upweighting of majority groups or minority examples, respectively, under group label noise. The authors show, among others, that under symmetric label noise both upweighting and downsampling lead to the same degradation of worst-group accuray.

Having shown the effect of group label noise, the authors propose a new approach which trains a linear model with high l1 penalty and uses the misclassifications to detect minority points. These are then upweighted to train the final model using a logistic loss with another l1 penalty.

The paper is empirically evaluated for CMNIST, CelebA, Waterbirds, MultiNLI and CivilComments by comparing the proposed approach against main competitor misclassification SELF (M-Self) and group-only upweighting and -downsampling (+ class-only methods,  vanilla last layer retraining in the appendix). The results show that the proposed approach is often superior to the baselines when more group label noise is added. The authors also show ablation studies of adding more group noise as well as increasing penalties during training.

**Audience:**

Yes

**Broader Impact Concerns:**

There are no concerns on ethical implications of the work.

**Claims And Evidence:**

No

**Requested Changes:**

- Please improve the presentation and discussion of the empirical results wrt class-only upweighting and downsampling. If this is mentioned as part of the evaluation, one should add the performances to the main line charts next to the other methods. Otherwise, one has to show tables of complete results in the main paper.

- Please clearly motivate the chosen main competitor misclassification SELF and resulting questions: Why can't one choose the best version of the respective paper, e.g., a disagreement variant? Would this use information you not allow in the problem statement? Otherwise I see no point in not comparing against this variant.  Lastly, if the overall goal of the paper is another, it should not be claimed that the approach beats the SotA under group label noise.

- Exending on that, it might be good to better explain how the proposed method differs to M-Self if one ignores the early stopping and drop out rates.

- Minor: It would be beneficial to add a short section or clear sentences what other techniques are currently investigated in recent papers, such as the mentioned SSL.

**Strengths And Weaknesses:**

# Strenghts
- Relevant problem setting: Worst-group accuracy optimization is an important topic, as spurious correlations are often learnt by deep learning models.
- Sensible proposed approach: The novel approach using high l1 penalization to identify minority samples is well-motivated.
- Empirical evaluation covers datasets similar to competitors: The chosen datasets seem sufficient to make claims
- Results show improvement of performance under group label noise against chosen competitors: Although the approach is not always the best compared to the baselines, it wins on average and show small variance over the runs.

# Weaknesses
- The empirical evaluation section is not convincing yet.
    - 1) The description states that the approach is evaluated against class-only upweighting/downsampling, but the results are not to be found in the main paper. They are only the the appendix and should be highlighted more.
    - 2) It seems the authors only compare to the worst version of LaBonte et al. (2023) with no clear argumentation. The exact method (M-Self) is mentioned directly, but why did the authors not compare against, for example, the early stop version or the best-performing disagreement version? It should be clearly stated, as this is the comparison of the State-of-the-Art. I understand that M-Self also uses misclassifications, which might be similar to the proposed approach using l1 loss, but if there are better strategies, why not bulid on these?
- Related work could give a better overview of other methods next to upweighting / downsampling. For example, recent methods  use SSL, such as Tsirigotis et al. (2023).
- Minor:
    - ERM is never introduced - only the abstract mentions empirical risk minimization but never refers to ERM.
    - In-sentence citations often miss surrounding brackts

References:
Tsirigotis, C., Monteiro, J., Rodriguez, P., Vazquez, D. and Courville, A.C., 2024. Group robust classification without any group information. Advances in Neural Information Processing Systems, 36.

---

> ### Author Response · Authors · 2024-09-02
>
> ### Rebuttal
> We thank the reviewer for their comments. We will address each of the presented concerns in turn, denoting weaknesses with W and requested changes with RC:
>
>  - **(W1.1/RC1)**	Class-only dependent methods are a logical baseline for this evaluation, but their performance is generally so poor as to distract from the comparison with more effective methods. For this reason, we chose to include the class-only results in the appendix. For clarity, we will mention this explicitly in the main body of the paper and include the performance when it does not distort the scale of the results. Additionally, we will mention the explicit values for each dataset in the discussion to aid the reader.
>
>  - **(W1.2/RC2)**	The comparison with M-SELF is presented because M-SELF is the only version presented by LaBonte, et al. (2023) which does not require side information, specifically early stopped versions of the base model. We assume access only to a fixed embedding function and a linear last layer, thus we restrict our comparison to only M-SELF. Utilizing the findings of LaBonte, et al. (2023) (e.g. disagreement) to minimize the number of required class annotations is an exciting avenue to explore, but is left as future work. We will clarify these constraints in the final version of the paper.
>
>  - **(RC3)**	Regarding the differences between our proposed method and M-SELF, the most important difference is that RAD does not reuse the existing last layer classifier, but instead trains a highly biased classifier using a strong L1 penalty. This allows the misclassifications to be more easily attributed to spurious features and results in a strong performance gain. An additional benefit to not reusing the pre-trained last layer is with regards to robustness. As pointed out in Iscen, et al., (2022) embeddings are generally fairly robust to noise in the upstream training/finetuning dataset, but the final classification head can exhibit drastically reduced performance. We have seen this in our experiments when testing with embeddings which were trained using noisy data. Consider an experiment where the neural network inducing our embedding function is trained using data with label noise. We see in the attached tables that both RAD and SELF suffer from noise in the embeddings, but SELF performs significantly worse as it also reuses the final classification head. We will add this experiment in full to the appendix to further clarify the benefits of RAD over SELF.
>
>  - **(W2/RC4)**	We appreciate the reviewers suggested related work and will ensure that our updated related work section includes discussion of various SSL-adjacent methods. We believe that RAD and other LLR methods still have promise as an efficient means of bias mitigation, and we hope that some of the advancements in the full-finetuning setting can be adapted to the LLR setting.
>
>  - **(W3)**	We will ensure that we clearly introduce ERM and fix these typos. Thank you!
>
> We hope that we were able to sufficiently address the concerns presented, and we believe that our additional results further emphasize the strength of RAD as compared to M-SELF.
>
> ### References
>
> A. Iscen, J. Valmadre, A. Arnab, and C. Schmid. Learning with neighbor consistency for noisy labels. In Proceedings of the IEEE/CVF Conference on Computer Vision and Pattern Recognition, pages 4672–4681, 2022
>
> ### Tables
>
> **CelebA**, embeddings trained with 20% SLN
> | Method | Clean Embeddings WGA | Noisy Embeddings WGA |
> |--------|----------------------|----------------------|
> | RAD    | 83.51 ± 0.02         | 80.55 ± 0.02         |
> | M-SELF   | 83.48 ± 0.02         | 31.86 ± 2.32         |
>
>
> **Waterbirds**, embeddings trained with 20% SLN
> | Method | Clean Embeddings WGA | Noisy Embeddings WGA |
> |--------|----------------------|----------------------|
> | RAD    | 92.52 ± 0            | 86.12 ± 0.05         |
> | M-SELF   | 92.83 ± 0.49         | 67.43 ± 4.48         |

---

### Review · Reviewer_RrSN · 2024-09-08

**Summary Of Contributions:**

Summary:
This paper studies the distribution robust optimization problem where there is subpopulation shift across different distributions, which is a very practical problem and could have potential applications in realistic situations. Specifically, there are different groups in training dataset, each of them is identified by a group annotation. However, in practice, the group annotation is normally not given. As a result, the difficulty of learning from groups of distributions is normally quite hard, and the performance on worst-case group is poor. Under this setting, the authors conduct thorough theoretical analysis to show that under symmetric group noise, the upweighting strategy and downweighting strategy achieve the same results. Further, the authors propose a novel method which simply uses a l1 regularization to ensure the performance of minor group data detection. Through extensive experimental evaluation, the performance of the proposed method is carefully justified.

**Audience:**

Yes

**Broader Impact Concerns:**

The author have stated broader impact, no ethical concerns.

**Claims And Evidence:**

Yes

**Requested Changes:**

Weaknesses:
- The proposed method shows too much similarity to existing methods, as the authors stated in the paper, many works have already leveraged l1 penalty to regularize the performance worst-case optimization, which makes the studies of this paper more of an incremental work. Please provide more justification on the novelty of this work.
- The experiments are quite extensive, various datasets and baseline methods are included in this paper. However, intuitive understanding of the proposed method is still unclear to me. For example, why not choosing the l2 or lp regularization? How will the strength of regularization affect the learning performance or group detection performance?
- There are several DRO references are missing:
  - Huang et al., Robust Generalization against Photon-Limited Corruptions via Worst-Case Sharpness Minimization, in CVPR 2023.
  - Indyk et al., Worst-case performance of popular approximate nearest neighbor search implementations: Guarantees and limitations, in NeurIPS 2023.
  - Chaudhuri et al., Why does throwing away data improve worst-group error? In ICML 2023.

**Strengths And Weaknesses:**

Strengths:
This paper is well written, the organization follows relatively clear logic and the methodology part is reasonable.
The theoretical contribution is intuitive and beneficial to the field of distribution robust optimization.
The experimental evaluations are quite exhaustive, which shows the effectiveness of the proposed method.

---

> ### Author Response · Authors · 2024-09-17
> **Rebuttal to Reviewer RrSN**
>
> We thank the reviewer for their feedback, and are especially grateful that the reviewer was pleased with our extensive evaluation of the proposed method. We would like to address each of the requested changes (RC):
>  - **(RC1)**	While existing methods have used L1 penalties in the retraining step (e.g. DFR), to our knowledge RAD is the first method to explicitly utilize the sparsity of the L1 solution for spurious sample identification, as opposed to core feature selection. The intuition, as discussed in Section 4 of the manuscript, is that the features in the latent space are made up of spurious, core, and junk features. Problems with fairness arise when spurious features are more strongly correlated with the label than true features. Thus, our identification step learns a sparse model based on spurious features. We want to only correctly label spurious examples, so that our error set (used for upweighting) consists of minority (non-spuriously correlated) examples for which the spurious features are not predictive. This is a reversal of the usual usage for an L1 penalty in this setting and sets our method apart from other two-stage methods.
>  - **(RC2)**	As for comparisons with other common regularizers, in Section 4 we compare against an L2 penalty for either the retraining or identification regularization. We find that using L1 in both the identification and retraining steps significantly increases the performance of RAD, especially on the Civil Comments dataset. L1 and L2 penalties are the focus because of their ease of optimization and intuition in this setting with regards to sparsity and overfitting.
>  - **(RC3)**	We thank the reviewer for their suggested references and will ensure that they are added to the related works section where relevant.
>
> We hope that this clarifies the novel intuition behind RAD and reinforces its benefits over existing methods.

---

### Decision · Action_Editor_2drE · 2024-11-04

**Recommendation:** Accept with minor revision

**Comment:**

After a thorough evaluation of the submission and the reviewers' feedback, I recommend accepting the paper with minor revisions. The paper makes a contribution by introducing a method that is both theoretically justified and empirically validated. The reviewers generally agree on the paper's strengths, particularly its clear problem statement, theoretical analysis, and extensive experimental evaluation.


Recommendations for Improvement:

- Enhance the discussion on how the proposed method differs from and improves upon methods like M-SELF. Emphasize the unique aspects, such as the use of L1 regularization for spurious feature identification and robustness to noise.

- Incorporate the experimental results comparing the proposed method with class-only upweighting/downsampling in the main paper to strengthen the empirical evidence.

- Expand the related work to include recent methods employing semi-supervised learning and other techniques relevant to worst-group accuracy optimization without group annotations.

- Address minor issues such as the introduction of ERM (Empirical Risk Minimization) and correct any citation formatting errors.

**Audience:**

This topic is highly relevant to the TMLR audience, which includes researchers and practitioners interested in robust machine learning, fairness, and domain adaptation.

**Claims And Evidence:**

The submission presents a new approach for optimizing worst-group accuracy (WGA) in the presence of spurious correlations and group label noise. The authors provide both theoretical analysis and empirical evidence to support their claims. They demonstrate that their method, which employs an explicit L1 regularization to identify minority samples and upweights them during training, outperforms state-of-the-art methods, especially under noisy conditions. The theoretical insights into the equivalence of upweighting and downsampling strategies under symmetric group noise are sound and add depth to the existing literature. The extensive experiments across multiple datasets validate the effectiveness of the proposed method, and the authors have adequately addressed the concerns raised by the reviewers.